# Environment and evolutionary history shape phylogenetic turnover in European tetrapods

Bianca Saladin [1], Wilfried Thuiller[2], Catherine H. Graham[1], Sébastien Lavergne[2], Luigi Maiorano [3], Nicolas Salamin [4,5] & Niklaus E. Zimmermann [1]

Phylogenetic turnover quantifies the evolutionary distance among species assemblages and is central to understanding the main drivers shaping biodiversity. It is affected both by geographic and environmental distance between sites. Therefore, analyzing phylogenetic turnover in environmental space requires removing the effect of geographic distance. Here, we apply a novel approach by deciphering phylogenetic turnover of European tetrapods in environmental space after removing geographic land distance effects. We demonstrate that phylogenetic turnover is strongly structured in environmental space, particularly in ectothermic tetrapods, and is well explained by macroecological characteristics such as niche size, species richness and relative phylogenetic diversity. In ectotherms, rather recent evolutionary processes were important in structuring phylogenetic turnover along environmental gradients. In contrast, early evolutionary processes had already shaped the current structure of phylogenetic turnover in endotherms. Our approach enables the disentangling of the idiosyncrasies of evolutionary processes such as the degree of niche conservatism and diversification rates in structuring biodiversity.

[1] Swiss Federal Research Institute WSL, 8903 Birmensdorf, Switzerland. [2] Univ. Grenoble Alpes, CNRS, Univ. Savoie Mont Blanc, LECA, Laboratoire d'Écologie Alpine, F-38000 Grenoble, France. [3] Dipartimento di Biologia e Biotechnologie 'Charles Darwin', Università di Roma 'La Sapienza', 00185 Roma, Italy. [4] Department of Computational Biology, University of Lausanne, 1015 Lausanne, Switzerland. [5] Swiss Institute of Bioinformatics, Quartier Sorge, 1015 Lausanne, Switzerland. Correspondence and requests for materials should be addressed to B.S. (email: bianca.saladin@wsl.ch)

The spatial structure of biodiversity has emerged through the interplay of ecological and evolutionary processes[1,2]. A better understanding of these processes is of utmost importance, especially as threats from global change increase, populations decline, and conservation decisions need to be taken[3,4]. Phylogenetic turnover (phylo-ß), which quantifies the phylogenetic distance among communities (i.e. pair-wise compositional turnover), can be used to explore the main drivers behind biodiversity patterns[5,6] and to define[7] or explain[1] the structuring of biodiversity in geographical space. Phylo-ß is affected by geographic and environmental distance between sites and represents macroecological and evolutionary processes of niche adaptation and separation by geographic barriers.

Only recently have studies begun to partition the effects of geographic and environmental distance on phylo-ß, a prerequisite for separating processes related to niche adaptation from those related to geographic dispersal barriers. Several studies using variation partitioning to tease apart the relative contribution of geographic and environmental distance found that either geographic[8] or environmental distance[9–11] predominantly explains phylo-ß among sites. Further, when the effect of geographic distance on phylo-ß was explicitly removed before analyzing phylo-ß along environmental distance, both low and high phylo-ß occurred at short and long environmental distances[8,12]. This unexplained variance suggests no clear relationship between phylo-ß and environmental distance (Fig. 1a). It can, thus, be hypothesized that phylo-ß is not a mere function of environmental distance but is instead strongly driven by the positioning along environmental gradients (such as temperature and moisture, see Fig. 1b, c). To our knowledge, no study has shown yet how phylo-ß is structured in environmental space.

Multiple evolutionary and ecological processes have shaped patterns of phylo-ß. These processes have driven the emergence of species richness and phylogenetic diversity and have also affected the niche size of species. Differences in lineage diversification along environmental gradients emerge from the interplay of speciation, extinction, dispersal capacity, and the associated shaping of ecological niches. For example, the magnitude of adaptive radiations may be constrained in extreme environmental domains (distinct regions within environmental space) due to costs of niche specialization (predicted by, for example, the theory of antagonistic pleiotropy or mutation accumulation)[13,14]. In contrast, less extreme environmental domains might promote higher diversification due to reduced costs of niche specialization that allow for easier radiations into neighboring habitats[13]. Therefore, different environmental domains may contain low or high species richness or phylogenetic diversity. These two diversity metrics are correlated and the deviation in phylogenetic diversity from the mean trend, termed relative phylogenetic diversity (the residuals of a regression explaining phylogenetic diversity based on species richness), can elucidate diversification processes[15]. Regions of high relative phylogenetic diversity are thought to have experienced high diversification rates of multiple lineages or immigration of multiple lineages that radiated successfully. In contrast, regions of low relative phylogenetic diversity may be associated with large radiations of few, closely-related lineages indicating that other lineages from the same taxon have not successfully colonized[15]. Therefore, trends in relative phylogenetic diversity along environmental gradients may strongly affect the emergence of phylo-ß patterns and point to the degree of niche conservatism. Evolutionary and ecological processes also affect the realized niche size of species. Evolution primarily affects the fundamental niche of species[16]. Yet, evolutionary adaptations in the competitive performance of a target species (e.g. through improved resource use efficiency) may also affect the realized niche of that species, and so does the adaptation of the

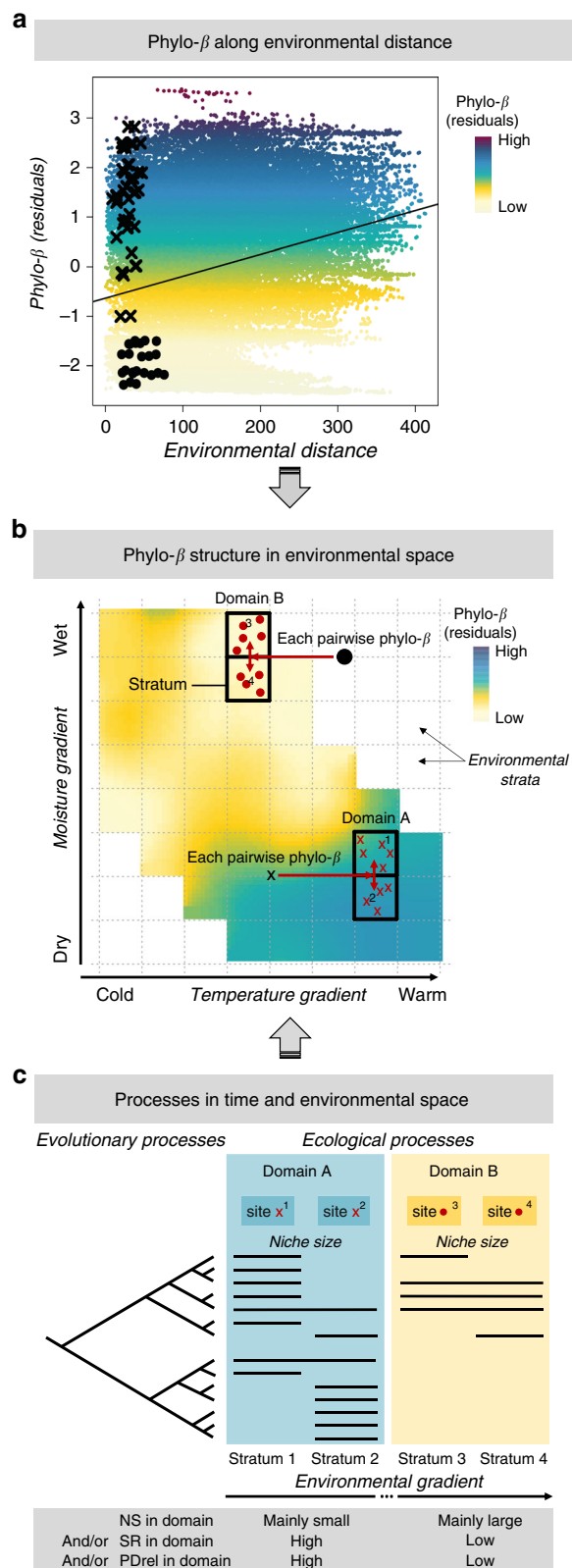

**a** Phylo-ß along environmental distance

**b** Phylo-ß structure in environmental space

**c** Processes in time and environmental space

fundamental niche of other, interacting species. Hereafter, niche size thus refers to the size of the realized niche available in a given region. Narrow niches enhance, while broad niches reduce phylo-ß along an environmental gradient. Therefore, changing niche sizes along an environmental gradient may strongly affect the emergence of phylo-ß patterns. In summary, we hypothesize that

**Fig. 1** Analysis and presentation of phylo-ß structure across environmental space. **a** Relationship between phylo-ß (corrected for geographical land distance, i.e. residuals of phylo-ß ~ land distance: phylo-ß (residuals)) and environmental distance among pairs of sample sites reveals large variation, indicating that distances are not necessarily good predictors of phylo-ß. Each black dot or cross refers to the mean of all pairwise phylo-ß calculations among sample sites (red dots or crosses in panel **b**) between neighboring strata. **b** Illustration of a hypothesized structuring of phylo-ß between neighboring equidistant strata (dotted and solid quadrats) resulting in environmental zones of low (domain B) or high (domain A) values. The mapped phylo-ß values (phylo-ß (residuals)) represent the means of all pairwise phylo-ß calculations among sample sites (red dots or crosses) between neighboring strata. The sample sites between neighboring strata share a relatively short environmental distance (see panel **a** for illustration), but originally had a large variation in geographic land distance (for which it had been corrected). **c** Illustration of possible macroecological drivers on phylo-ß among sample sites (e.g. sites 1, 2 for domain A; sites 3, 4 for domain B). Phylo-ß is hypothesized to be explained jointly (and relation) or independently (or relation) by niche size (NS), species richness (SR), relative phylogenetic diversity (PDrel)

these three macroecological characteristics (species richness, relative phylogenetic diversity, and niche size) can jointly or independently explain variation in phylo-ß (Fig. 1c).

Here, we quantify and decipher the patterns of phylo-ß in environmental space for European tetrapods. We do so by analyzing phylo-ß patterns (based on true turnover[17,18]) across European samples of tetrapod distributions, and by mapping phylo-ß in equidistant steps (environmental strata, hereafter) along environmental gradients (Fig. 1). Our analyses are carried out separately for the four major clades of European tetrapods (i.e. amphibians, birds, mammals, and squamates). These clades are particularly well suited for testing our hypotheses because the two major groups (ecto- and endotherms) differ in their thermal physiology, niche evolutionary rates, climatic tolerances, and dispersal capacities[19], leading to distinct broad-scale ecological patterns. The novelty of our approach is twofold. First, we map phylo-ß across environmental space after having removed the effect of geographic land distance among sample sites on phylo-ß (see Fig. 1 and Methods), rather than testing distance decays or to what degree phylo-ß can be partitioned into geographic and environmental distance effects. Second, we seek to explain the resulting phylo-ß patterns based on macroevolutionary characteristics to assess the different processes shaping phylo-ß.

Specifically, we find that: (1) phylo-ß is structured in environmental space, (2) the patterns of phylo-ß in environmental space are stronger when emphasizing recent rather than deep branches in ectotherms, (3) phylo-ß patterns are stronger in ectotherms than in endotherms, and (4) macroecological characteristics well explain phylo-ß in environmental space, though with differing predictive power among clades. These results unravel distinct evolutionary processes among the four clades shaping their biodiversity patterns in environmental space.

## Results
**True phylogenetic turnover across geographic land distance.** Geographic land distance explained some of the variation in true phylogenetic turnover (Simpsons pairwise dissimilarity index) between pairs of sample sites (Supplementary Figure 1a; adjusted regression $R^2$ for amphibians: 0.03, squamates: 0.04, mammals: 0.29 and birds: 0.46). Hereafter, true phylogenetic turnover (phylo-ß) refers to the fraction that is independent of geographic land distance (residuals of Simpsons pairwise dissimilarity index ~ geographic land distance).

**True phylogenetic turnover across environmental distance.** Phylo-ß varied greatly across environmental distances (Supplementary Figure 1b). Pearson correlations between phylo-ß and environmental distance were significant but relatively low across all clades (amphibians: $r = 0.33$, 95% CI [0.327, 0.342], $p < 0.001$; squamates: $r = 0.36$, 95% CI [0.352, 0.371], $p < 0.001$; mammals: $r = 0.29$, 95% CI [0.279, 0.294], $p < 0.001$; birds: $r = 0.35$, 95% CI [0.345, 0.359], $p < 0.001$). The variation in phylo-ß was especially high among short environmental distances, irrespective of the clade analyzed. To visualize the structure in and the drivers behind this variation, we plotted phylo-ß along short, equidistant neighboring strata (Fig. 2).

**Patterns of phylogenetic turnover in environmental space.** Mean phylogenetic turnover (phylo-ß) between sites of neighboring environmental strata was strongly structured in environmental space defined by temperature and moisture after having removed the effect of geographic distance between sites on phylo-ß (Fig. 2). Environmental structuring of phylo-ß was more robust in ectotherms than in endotherms when the random sampling of communities was repeated (Supplementary Figure 2) and when using PCA axes instead of explicit climate gradients (Supplementary Figure 3 and Supplementary Table 1, for additional robustness analyses see Methods). The two ectotherm groups revealed a similar pattern with high phylo-ß in drier and warmer environmental domains and low phylo-ß in colder and wetter domains, although phylo-ß increased in amphibians towards very cold and wet sites. Thus, a clear difference was found between dry-warm and cold-wet conditions. In contrast, for the endotherms, the barrier between high and low phylo-ß was rather from cold-dry domains (high phylo-ß in mammals, low phylo-ß in birds) towards warm-wet domains (low phylo-ß in mammals, high phylo-ß in birds).

Emphasizing very recent ($\delta = 10$), relatively recent ($\delta = 3$), or deeper phylogenetic branches ($\delta = 0.3$ and $0.1$) when calculating phylo-ß revealed differences among the four taxonomic groups. Compared to the endotherms, ectotherms revealed stronger gradual changes in the phylo-ß structure through evolutionary time: phylo-ß patterns were weak when emphasizing deeper branches and became stronger when emphasizing recent branches (Fig. 3 and Supplementary Figure 4 for $\delta$-values 0.1 and 10). In contrast, phylo-ß patterns in endotherms became only marginally stronger with higher $\delta$-values. These results are similar to what is found when truncating the tree at certain depths in the phylogeny rather than $\delta$-transforming the branches (see Supplementary Figure 5).

**Phylogenetic turnover and macroecological characteristics.** Species richness, relative phylogenetic diversity, and niche size were also distinctly structured in environmental space (Supplementary Figure 6). Species richness patterns for amphibians and mammals were very similar, with high richness across most environmental conditions in Europe, although somewhat lower in cold-moist conditions for amphibians. In squamates, species richness was high in warm domains and rapidly diminished towards cooler conditions. Species richness in birds peaked at intermediate values of moisture and temperature and declined towards cold-humid and hot-dry domains. Relative phylogenetic diversity was clearly structured in amphibians, mammals, and birds, with a tendency towards negative relative phylogenetic diversity values in cold-wet domains. Relative phylogenetic diversity for squamates was negative in comparably hot and comparably cold conditions and positive in-between, and overall positive in very warm conditions. The ectotherm clades showed lowest niche size values in dry-warm domains and higher values towards cool-wet domains, while mammals had

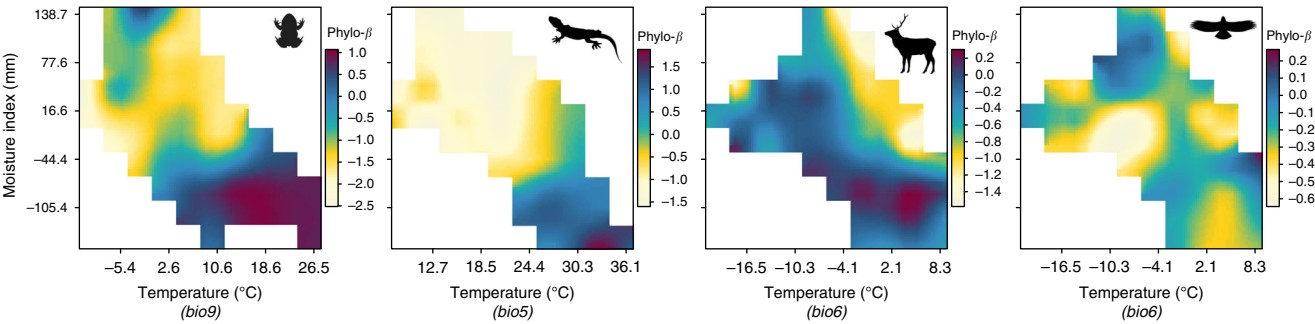

**Fig. 2** True phylogenetic turnover in environmental space. Phylo-ß is illustrated for amphibians, squamates, mammals, and birds (from left to right) and is corrected for geographic land distance between sites (see Method). Environmental space is defined by moisture index and temperature (represented by bioclim variables[60], selected for each clade based on their highest explanatory power: bio9 = mean temperature of driest quarter, bio5 = maximum temperature of the warmest month, bio6 = minimum temperature of the coldest month). Silhouette images were taken unchanged from phylopic.org, courtesy of Pedro de Siracusa (amphibians), Ghedo and T. Michael Keesey (squamates) both available under CC BY-SA 3.0 (creativecommons.org/licenses/by-sa/3.0/), Steven Traver (mammals) available under CC0 1.0 (creativecommons.org/publicdomain/zero/1.0/), and Shyamal (birds) available under CC BY 3.0 (creativecommons.org/licenses/by/3.0/)

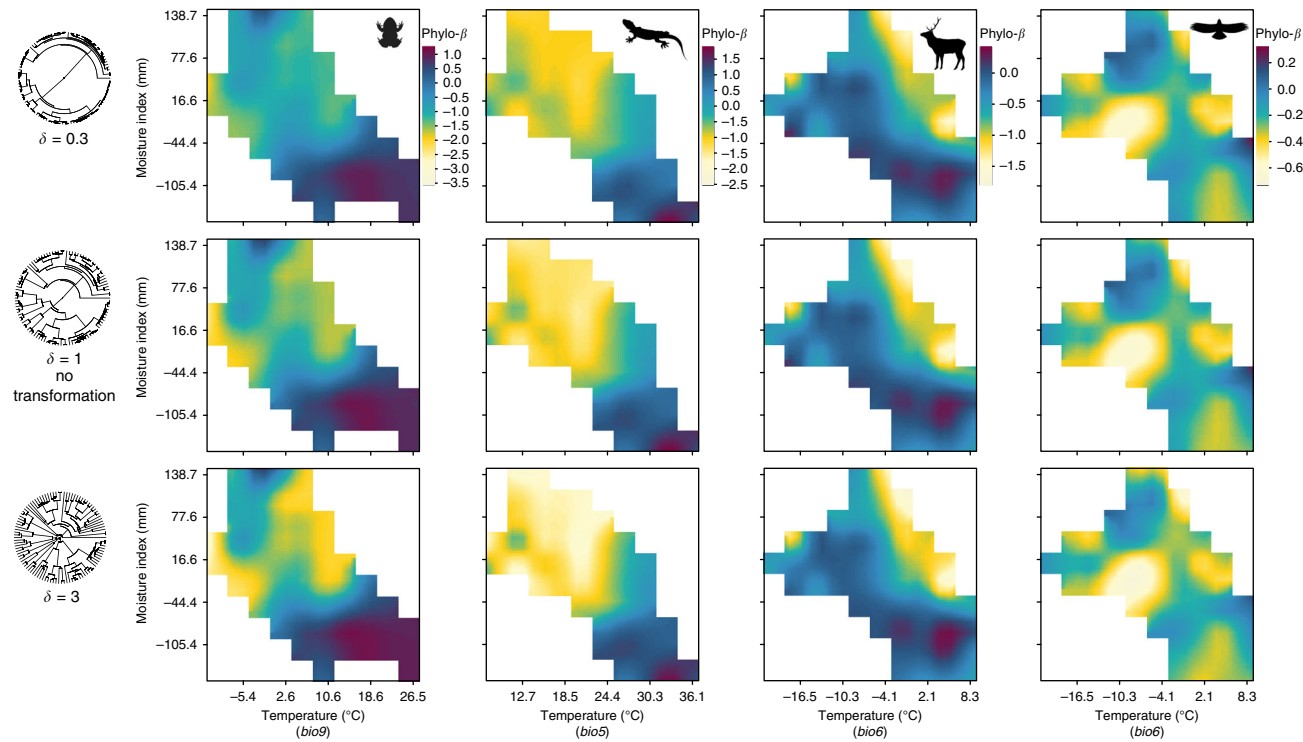

**Fig. 3** True phylogenetic turnover in environmental space using rescaled phylogenetic trees. Phylo-ß is represented for amphibians (1st row), squamates (2nd row), mammals (3rd row), and birds (4th row) and is corrected for geographic land distance between sites. Rescaling of trees was based on delta transformation (δ > 1 emphasizes recent branches; δ < 1 emphasizes deep branches). Note: for each clade, the same color range was applied to illustrate phylo-ß for comparison (therefore, the colors for δ = 1 do not exactly match with Fig. 2). Silhouette images were taken unchanged from phylopic.org, courtesy of Pedro de Siracusa (amphibians), Ghedo and T. Michael Keesey (squamates) both available under CC BY-SA 3.0 (creativecommons.org/licenses/by-sa/3.0/), Steven Traver (mammals) available under CC0 1.0 (creativecommons.org/publicdomain/zero/1.0/), and Shyamal (birds) available under CC BY 3.0 (creativecommons.org/licenses/by/3.0/)

higher niche size values towards warm-wet domains. Niche size values of birds were highly homogenous across the environmental space of Europe.

The relative importance of these macroecological characteristics in explaining phylo-ß differed among the four clades of European tetrapods (Fig. 4) and the linear model fit was considerably higher for ectotherms ($R^2$: amphibians = 0.70, squamates = 0.79) than for endotherms ($R^2$: birds = 0.20, mammals = 0.46). In amphibians, squamates and birds, phylo-ß was best explained by trends in species richness (46–72% in relative

importance), whereas in mammals, niche size had the highest importance (72%). In amphibians, relative phylogenetic diversity was more important in explaining phylo-ß (47%) than niche size (6%), the opposite was found for squamates where niche size explained more (37%) than relative phylogenetic diversity (7%).

## Discussion
We successfully decipher the imprints of macroecological characteristics indicative of evolutionary processes on patterns of

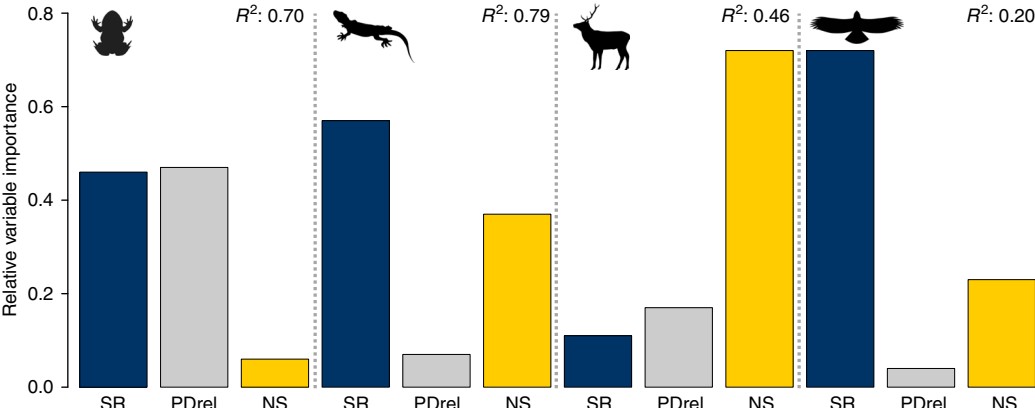

**Fig. 4** Relative variable importance of macroecological characteristics. Variable importance of species richness (SR), relative phylogenetic diversity (PDrel), and niche size (NS) in explaining phylo-ß in environmental space is given for amphibians, squamates, mammals, and birds. $R^2$ represents the model fit of a standardized linear regression to explain phylo-ß by means of standardized SR, PDrel, and NS (linear and quadratic terms). Relative variable importance values are re-scaled to add up to sum of 1.0. Silhouette images were taken unchanged from phylopic.org, courtesy of Pedro de Siracusa (amphibians), Ghedo and T. Michael Keesey (squamates) both available under CC BY-SA 3.0 (creativecommons.org/licenses/by-sa/3.0/), Steven Traver (mammals) available under CC0 1.0 (creativecommons.org/publicdomain/zero/1.0/), and Shyamal (birds) available under CC BY 3.0 (creativecommons.org/licenses/by/3.0/)

phylo-ß in four taxonomic groups at continental scale. First and as expected, we find a distinct structuring of phylo-ß in environmental space after removing the effect of geographic land distance, indicating the constraining effect of environmental conditions on the evolution of niches and the emergence of modern biodiversity patterns. Second, this structure is influenced by evolutionary processes that have driven phylo-ß patterns at different times in the evolutionary history of the four taxonomic clades as indicated by a varying influence of δ-transformations on phylo-ß structures among clades. Third, the phylo-ß structure varies among the four tetrapod clades and is generally stronger and more robust in ecto- than in endotherms. This indicates that the underlying evolutionary processes that have shaped biodiversity turnover along environmental gradients have been generally different for ecto- and endotherms, as is evidenced by the explanatory power of species richness and relative phylogenetic diversity and of occupied niche sizes in environmental space when explaining phylo-ß. Our results are robust regarding the analysis choices made, as changing the spatial resolution of initial range maps (Supplementary Figure 3), the number of environmental strata used (Supplementary Figure 7) and the choice of phylogenetic trees (Supplementary Figures 8 - 11) did not change the general phylo-ß structure.

The discontinuities in phylo-ß we found along environmental gradients are generally congruent with the view that phylo-ß is not strongly correlated with environmental distances, a common pattern found in empirical studies[5,20,21] and in our data (Supplementary Figure 1b). Even though variation in phylo-ß along an elevation gradient has been found, e.g. in plants[21], it was unexplored whether discontinuities remain after removing the effect of geographic distance. Elevation gradients do not represent environmental distances linearly since elevation is usually only indirectly related to other environmental variables[22] and they differ considerably in warm and cold environments[23]. More generally, spatial gradients can reveal strong structuring in phylo-ß, as was demonstrated by Peixoto et al.[24]. Yet the latter study did not separate the effects of geographic from environmental distance, and therefore the main drivers behind the found patterns could not easily be addressed. If the focus is on environmental drivers, it is important to interpret biological patterns directly against measured environmental gradients rather than against geographical clines, i.e. as a function of latitude or altitude[22,25,26].

To our knowledge, Penone et al.[12] were the first to remove the effect of geographic distance on phylo-ß in their global analysis of mammals and indeed, they found high variation in phylo-ß along a gradient of environmental distance, yet environmental distance as such did not explain the remaining variation. Our results indicate that it is not simply environmental distance that explains patterns in phylo-ß. Rather, it is the positioning within an explicit environmental framework that explains high vs. low phylo-ß values, as was hypothesized earlier by Warren et al.[27] from a conceptual perspective and supported by the relationship found between phylogenetic turnover and variation in local environmental conditions by Rosauer et al.[28]. However, phylo-GDM calculates turnover among all sample sites. Therefore, the phylo-ß patterns found are not explicitly analyzed for equidistant progression along environmental gradients as in our study and may thus tend to average phylo-ß – environment relationship over various environmental extents. Our method is a step forward in the analysis of phylo-ß patterns and an implementation of the quest by Warren et al.[27] as it explicitly reveals phylo-ß patterns in environmental space. As a result, our approach allows us to focus on environmental drivers (environmental filtering or evolutionary niche adaptation processes) and avoids overestimation of, or confusion with, geographic drivers (barriers, limited migration, and allopatric speciation) in explaining phylo-ß. This represents an important advance in our understanding of the spatial structure of biodiversity.

The relationship between phylo-ß and the environment has been shown to peak at particular depths in a phylogenetic tree[27,28]. For instance, the relationship was found to be strongest for recent phylogenetic relationships in myobatrachid frogs and was confounded when emphasizing deeper branches by cutting the tree at different time depths[28]. Here, as proposed by Chalmandrier et al.[29], we used δ-transformed phylogenies to analyze at what depth the phylo-ß structure in environmental space was strongest, and we found different clades to show strongest phylo-ß structuring at different δ-values (Fig. 3 and Supplementary Figure 4). For European ectotherms, we found that emphasizing recent branches strengthens these patterns considerably, while weighting deeper branches weakens them. This indicates that rather recent evolutionary processes were important in structuring the observed phylo-ß pattern along environmental gradients. In comparison, phylo-ß structures in mammals and birds along

environmental gradients were driven by evolutionary processes that acted earlier in the history of the group, which suggests that several older clades have radiated within distinct environments[28]. These results emphasize the importance of considering phylogenetic scale[30].

The ecological and evolutionary processes differ between ecto- and endotherms along the analyzed environmental gradients as indicated by: (a) the more distinct and robust patterns of phylo-ß in environmental space for ecto- than for endotherms (Fig. 2), and (b) the difference in explanatory power (Fig. 4) of macroecological characteristics between the different clades in explaining phylo-ß. This likely originates from differences in their life strategies: ectotherms generally have smaller geographical ranges[31], narrower environmental niches[19,32] (but see niche size of amphibians in Europe, Supplementary Figure 6), are ecologically more constrained in their capacity to survive (primarily under cold environments[33]), and are more limited in their dispersal[34] than mammals and especially birds. Moreover, climatic niches in endotherms have evolved more rapidly than those of ectotherms[19], likely influencing species richness and lineage diversification[35]. Endotherms also showed significantly faster niche shifts than ectotherms[19], promoting phylogenetic lability of niches. Thus, the overall phylogenetic turnover is expected to be steadier (with no clear peaks in turnover) and higher along environmental gradients in endo- than in ectotherms. This is in agreement with the less clear and less robust environmental transition zones of phylo-ß found in endotherms in our analyses, with comparably high phylo-ß values throughout most of the analyzed environmental domain. In contrast, the transition zones found in phylo-ß in ectotherms may indicate an environmental constraint on diversification towards cold-wet climatic domains, especially in amphibians with only few (low species richness) closely related lineages (negative relative phylogenetic diversity) that have adapted to this domain with comparably large niches (see Supplementary Figure 6). Compared to amphibians, the phylogenetic structure was less pronounced in squamates, meaning that the few clades that have adapted to cold-wet domains are less closely related than those in amphibians (e.g. indicated by the low importance of the relative phylogenetic diversity in explaining phylo-ß patterns). This indicates weaker niche conservatism or slightly faster rate of niche shifts in squamates than in amphibians (at least in European species). The patterns found may be influenced by the geographic context of Europe, where only a part of the global niche space of tetrapods is available.

Our analyses do not capture all possible drivers that may have shaped phylo-ß in environmental space. For example, historical tectonic and climatic dynamics may have also left an imprint on phylo-ß along environmental gradients. Particularly Pleistocene glaciations have likely affected the diversity patterns of European organisms through processes of range contractions, range expansions, and unequal extinction in environmental space[36–38]. Furthermore, richness of herptile species is best explained by past glacial refugia rather than that of current climate[39]. Relying only on living species and current climate to understand the patterns of phylo-ß and niche evolution could thus partly be misleading[40]. The inclusion of fossils and past climate environment data are candidates for improving our understanding of modern phylo-ß patterns in environmental space. In addition, evolutionary process models[41] and specifically-designed experiments[13] can additionally be used to test hypotheses that emerge from the analyses of phylo-ß patterns in environmental space.

Our results clearly show that phylo-ß varies across environmental space and that this variation can be explained by an interplay of several macroecological characteristics. Only a combination of these characteristics can give insights into the

mechanisms that drive the evolution of biodiversity in response to environmental constraints. Patterns in species richness, relative phylogenetic diversity and niche size give insights into evolutionary and ecological processes, while phylo-ß emphasizes and localize constraints to these processes in environmental space. Extending this study to regions that include (sub-) tropical climates would allow for a more complete picture of the turnover patterns in tetrapods, as their physiological limits are not fully reached in Europe.

## Methods

**Analysis pathway and data preparation**. The main analysis steps are shown in Supplementary Figure 12. The study area spans the entire European subcontinent, except smaller and less connected Islands (Iceland, parts of Russia, Balearics, Corsica, Sardinia) as their evolutionary history might be highly biased by limited dispersal, extinction history, and local evolutionary constraints. We further excluded parts of Eastern Europe due to a lack of quality in occurrence data in those regions (Supplementary Figure 2a). We only included raster cells that were fully covered by terrestrial land to avoid sampling bias in subsequent analyses from area-richness effects.

We used distribution maps compiled by Maiorano et al.[42] and up-scaled them from a 300 m to a 15 km resolution using the raster package in R[43]. To work with presence and absence data, we scored the originally distinguished primary and secondary habitat as presence and unsuitable habitat as absence. Within our study area, a total of 67 amphibians, 95 squamates, 161 mammals, and 414 birds were present.

We used the dated species level phylogenetic trees inferred by Roquet et al.[44] and available on Dryad[45]. Therein, the phylogenetic trees for amphibians and squamates were built from gene sequences downloaded from Genbank for the European species included in ref. [42], which contain 102 amphibians and 236 squamates. The phylogenetic trees for amphibians were based on 9 mitochondrial (*12 S, 16 S, COI, cytb, ND1, ND2, ND4, tRNA-Leu,* and *tRNA-Val*) and 2 nuclear (*RAG-1, rho*) regions and were all available for at least 30% of the species. For European Squamata, the phylogenetic inference was based on 7 nuclear (*BDNF, c-mos, NT3, PCD, R35, RAG-1,* and *RAG-2*) and 6 mitochondrial loci (*12 S, 16 S, COI, cytB, ND2,* and *ND4*). Only 16 species out of 239 squamata species had no molecular data available and were placed based on taxonomic knowledge. The phylogenetic trees for birds from Roquet et al.[45] were originally from ref. [46], include 430 species of European breeding birds, and were built upon 10 mitochondrial gene regions (*12 S, ATP6, COII, COIII, ND1, ND2, ND4, ND5,* and *ND6*) and 6 nuclear regions (*28 S, c-mos, c-myc, RAG-1, RAG-2,* and *ZENK*). The phylogenetic trees for mammals were based on the super-tree of Fritz et al.[47], which originally is based on 34 mitochondrial and 32 nucelar genes distributed across a total of 2182 species (out of a total of 4510 species). The mammal trees we derived from Roquet et al.[44] were generated by extracting 100 fully resolved phylogenetic trees by Kuhn et al.[48] who randomly resolved the polytomies of the super-tree of Fritz et al.[47] using a birth-death model to estimate branch-lengths. Roquet et al.[44] then additionally replaced the Carnivora clade with an update by Nyakatura and Bininda-Emonds[49] and removed all non-European species. For amphibians, squamates and birds, phylogenetic inference was conducted with RaxML searching for 100 Maximum Likelihood trees applying a family tree constraint (for amphibians based on Roelants et al.[50], for squamates on Pyron et al.[51] and for birds on Hacket et al.[52]). The trees for these three groups were dated using penalized-likelihood (r8s) based on fossil information (in amphibians 7 selected nodes were constrained; in squamates 5 nodes, in birds 14 nodes, for further details see ref. [44]). We checked for concordance in names between tips of the phylogenetic trees and species of which we had distribution data. This resulted in 59 amphibians, 94 squamates, 157 mammals, and 384 birds remaining in the analyses (Supplementary Figure 8 - 11). We randomly selected 20 phylogenetic trees from each set of tree source for our analyses. The phylogenetic trees for amphibians and squamates were built on the set of European species analyzed in the study. The phylogenetic trees for mammals and birds were sub-sampled from a global representation of the species. As this difference in generating phylogenetic trees could affect our results, we also ran the analyses by using a sub-sampled global phylogenetic tree for amphibians and squamates. See section on robustness tests regarding possible effects of the choice of the phylogenetic source.

To assess whether the found patterns are related to rather recent or rather deep lineages, we rescaled the phylogenetic trees based on the δ-transformation[53] using the Geiger package in R[54], similar to Chalmandrier et al.[29]. This transformation changes the distances of branches: δ > 1 disproportionally increase the length of recent branches, while δ < 1 disproportionally increase the length of deep branches. We chose δ-value of 0.1, 0.3, 1.0 ( = no transformation), 3 and 10. An alternative method of this analysis step would be to truncate the phylogeny at certain phylogenetic depths, as developed[55] and implemented[56,57] in recent studies. For reasons of comparison, we also implemented this method. We pruned the phylogenies at five specific time periods (depths) and collapsed all descendant leaves of each of the branches encountered by the pruning (see Supplementary

Figure 5). The geographical distribution of these branches is defined as the union of the distributions of their descending leaves.

We chose two environmental variables that represent important climate constraints on species distributions, niche evolution, and adaptations. Specifically, they reflect adaptations to extreme cold or extreme drought. For measures of drought, we used the mean moisture index of the three warmest months (June, July, and August). This index represents the difference (in mm) of monthly precipitation and potential evapotranspiration as calculated based on the Turc equation[58] using means of monthly average temperature and solar radiation. All three required monthly climate layers were taken from Worldclim2[59] at a 1 km resolution and were then up-scaled to a 15 km resolution to match the distribution data. For temperature, we selected for each taxon group a variable tested for highest explanatory power for phylo-ß, which gave us: mean temperature of the driest quarter (amphibians), maximum temperature of the warmest month (squamates), and minimum temperature of the coldest month (birds and mammals). We downloaded these variables from Worldclim[60] at a 1 km resolution and up-scaled them to a 15 km resolution. In order to perform a stratified random sampling of the distribution data across Europe for analyzing diversity (alpha) and turnover (beta) measures, we stratified both environmental raster layers (temperature and moisture index) into 9 equally sized classes and combined them to one environmental stratification layer consisting of 81 possible strata (see section on robustness test to evaluate the effect of the number of strata). This group-specific selection of bioclim variables allows for a better ecological interpretation than when using PCA axes from many climate variables. Yet, we also used PCA axes to test the robustness of our analytical procedure (see section on robustness test).

**Stratified random sampling.** Using the stratification layer, we allocated sample sites geographically at random within each environmental stratum for each taxon group separately. We present the results from one of the five sets of sample sites in the main text (see section on robustness tests for effects of repeated sampling allocation). The number of sampled sites within a stratum was set proportional to the log of the number of pixels per stratum with a maximum of 10 (in the largest stratum) and a minimum of 1 (in the rarest stratum). We chose a maximum of 10 random sites for computational efficiency reasons. No sample sites were allocated to locations within strata where no species occurred. We obtained 322 sample sites for amphibians, 253 for squamata, 326 for mammals, and 326 for birds, respectively. For each sample point, we recorded the presence and absence of each species for further analyses.

**Phylogenetic turnover and macroecological characteristics.** To calculate phylogenetic turnover (phylo-ß) between neighboring strata, we grouped all sampled sites per stratum and calculated mean phylo-ß from all pairwise dissimilarity calculations (Simpsons pairwise dissimilarity index) of sample sites between the two strata using the betapart package[61] in R. To remove the effect of geographic distance on phylo-ß we performed a generalized linear model with logit-transformed phylo-ß as dependent variable and geographic distance (linear and quadratic terms) between sampled sites as explanatory variable. Geographic distance is calculated here as land distance, thus penalizing trajectories through sea by a factor of 10 (using a least-cost path distance analysis with 10x higher costs over sea compared to land). Robustness analyses using both an even higher sea trajectory penalty (100×) and using linear (Euclidean) distance with no penalty for sea trajectories are given in Supplementary Figure 13. The residuals from the regression represent the phylo-ß fraction that is independent of the land distance among sampled sites. This approach is similar as the one used in previous studies[8,12]. It represents an element of variation partitioning[62] and removes the geographic distance effect and also some fraction of the environmental distance effect, as the two effects are not fully independent. However, in phylo-ß analyses of continental scale, geographic and environmental distances usually only explain a small proportion and the unexplained fraction is usually very high[8,12]. Yet, our approach allows for mapping both the environmentally structured and the unexplained fraction directly in to environmental space in order to interpret the results in a direct ecological context, and not simply in a distance context. An alternative approach for removing the effect of geographic distance would be to use eigenvector-based techniques[63]. To consider phylogenetic uncertainty, we calculated phylo-ß using a set of 20 randomly chosen phylogenies and calculated the mean thereof for further analyses. All the steps of the analyses were repeated for the five δ-transformations separately.

To explain phylo-ß from macroecological measures of diversity and niche size, we first calculated species richness and phylogenetic diversity. We then calculated the relative phylogenetic diversity, which represents the deviation (residuals) from a linear model (with linear and quadratic terms remaining significant) of species richness explaining phylogenetic diversity[15]. Relative phylogenetic diversity expresses unusually high (positive) and unusually low (negative) deviations of phylogenetic diversity given the underlying species richness. That is, high relative phylogenetic diversity highlights areas of co-occurring taxa from distantly related lineages or it represents species rich areas due to high immigration rates of multiple lineages[15]. In contrast, low relative phylogenetic diversity relates to radiations of few closely related lineages[15]. The size of the realized niche was calculated as bivariate ellipse area along the two niche axes used in this analysis (temperature and moisture index). Ellipse areas were set to encompass 95% of each species' data

points when mapping their ranges onto these two niche axes and calculations were performed using the car package[64] in R. Our calculations may underestimate the niche size of species that also occur outside Europe.

**Interpolation into environmental space.** We interpolated phylo-ß (that is the residual values that represent the phylo-ß fraction that is independent of varying geographic land distance between sample sites), species richness, relative phylogenetic diversity, and niche size into environmental space defined by the temperature and moisture gradients. To do so, we rasterized each variable in the raster package[43] using bilinear interpolation.

**Variable importance for explaining phylo-ß.** We used standardized linear regression models to explain phylo-ß based on (standardized) species richness, relative phylogenetic diversity, and niche size (linear and quadratic terms) and to quantify the variable importance in the phylo-ß models for each taxon group separately. Model fit was reported as model $R^2$ and variable importance was calculated based on the lmg method implemented in the relaimpo package[65]. Importance values were re-scaled to sum up to 100%, importance values from linear and quadratic terms were added up per variable, and finally plotted as barplot by taxon group.

**Robustness analyses.** To assess the robustness of our results, we ran the same analyses by using: (a) a different set of phylogenetic trees from another source; (b) different spatial resolutions of the distribution data; (c) a PCA of 13 climate variables instead of two major climate variables; (d) different numbers of climatic strata (resolution of stratification layers); and (e) four additional repeats of the stratified random sampling. For (a), we used the phylogenetic trees of major clades from the evolutionary timetree of life (TTOL) of Hedges et al.[66] (downloaded from http://www.biodiversitycenter.org/ttol), which is based on assembled timetree data from numerous published studies between 1987 and April 2013. For birds and mammals, two kinds of phylogenetic trees were provided and used in our study (unsmoothed and smoothed-interpolated), while for amphibians and squamates only an unsmoothed version was provided and used. The unsmoothed phylogenetic tree only includes species for which molecular data was available and polytomies were not randomly resolved. The smoothed and interpolated phylogenetic tree additionally includes species lacking molecular data, which were phylogenetically placed based on taxonomic knowledge and its polytomies were resolved by random placement of branches within the specified taxonomic group. We checked for concordance in names between tips of the phylogenetic trees and species of which we had distribution data. This resulted in 62 amphibians, 83 squamata, 158 mammals (smoothed), 153 mammals (unsmoothed), 387 birds (smoothed), and 376 birds (unsmoothed) remaining for analyses (Supplementary Figure 8 – 11). In (b), we reprojected the original distribution data to resolutions of 9 km and 21 km. In (c), we ran the analyses with the first two axes of a PCA of 12 bioclimatic variables[60] and the moisture index. We chose six temperature variables (bioclim variables 2, 4, 5, 6, 8, and 9) and six precipitation variables (bioclim variables 12, 15, 16, 17, 18, and 19) that show low correlations across Europe. In (d), we stratified the climatic layers into 5, 13, and 17 classes, each. In (e), we additionally ran the phylo-ß analyses and interpolations using the remaining 4 sets of stratified random sampling sites.

**Code availability.** All relevant R code is available from the Dryad Digital Repository: https://doi.org/10.5061/dryad.sr7818c [10.5061/dryad.sr7818c][67].

**Reporting summary.** Further information on experimental design is available in the Nature Research Reporting Summary linked to this article.

## Data availability
The source data underlying Figs 2–4 and Supplementary Figs 1-11 and 13 are provided as a Source Data file, which is available from the Dryad Digital Repository: https://doi.org/10.5061/dryad.sr7818c [10.5061/dryad.sr7818c][67]. The original tetrapod phylogenies are available from: https://doi.org/10.5061/dryad.11609 [10.5061/dryad.11609][45] and the phylogenies for Supplementary Figs 10-11 are available from the Center for Biodiversity [http://www.biodiversitycenter.org/ttol]. A reporting summary for this Article is available as a Supplementary Information file.

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

## Acknowledgements
This study was supported by the Swiss National Science Foundation (Grant 31003A_149508/1). We would like to thank Rafael O Wüest for helpful discussions. We thank the following persons for silhouette images: Pedro de Siracusa (amphibians), Ghedo, and T. Michael Keesey (squamates) under CC BY-SA 3.0 (creativecommons.org/licenses/by-sa/3.0/), Shyamal (birds) under CC BY 3.0 (creativecommons.org/licenses/by/3.0/) and Steven Traver (mammals) under CC0 1.0.

## Author contributions
B.S. and N.E.Z. conceived the idea with additional input from W.T.; L.M. provided distribution data. B.S. performed the analyses, B.S. and N.E.Z. conceived the figures. B.S. wrote the first draft of the manuscript and B.S., W.T., C.H.G., S.L., L.M., N.S., and N.E.Z. contributed substantially to subsequent revisions.

## Additional information

**Competing interests:** The authors declare no competing interests.

