## [Peer Review File · Nature Communications]

Reviewers' Comments:

Reviewer #1:

Remarks to the Author:

This manuscript examines the environmental correlates of phylogenetic beta diversity for 4 major tetrapod radiations in Europe. The key innovation of the study is to control for the component of beta diversity which is explained by geographic distance in order to reveal the component of compositional turnover which is specific to differences in environment.

The paper is well written, with thorough and interesting analysis. The rationale and findings of the study are well argued.

There are several concerns which it would be important to address. None of these are fatal, but they may require some reanalysis or changes to framing of the problem and results.

1) The method used, evaluating phylogenetic turnover (phylo beta) in environmental rather than geographic space, is an effective way to remove the effects of geographic distance. But can the resulting phylo beta patterns can be interpreted as representing correlates of environment alone? Geographic distance is important to beta diversity as a surrogate for connectivity or isolation between communities. But connectivity depends on what lies between two locations, not just on the distance. For example, Italy and the Balkans are geographically close, but separated by the Adriatic Sea, while a similar distance in another part of Europe may have few significant barriers. And of course there are the Alps as a barrier (and corridor). So, the patterns which the ms presents as purely about environment, could be expected to actually reflect both environment, and connectivity.

For example could this help explain the higher turnover found in warmer environments (Fig 2 - amphibians, squamates). This is presented in terms of temperature, but could relate to sea barriers between major land areas of southern Europe, which contribute to higher phylogenetic turnover per unit of distance. The problem is complex, but the authors need to be more nuanced about the study's ability to "evaluate phylo beta in a purely environmental context" (lines 49-50). One option to consider would be to use land distance between cells in place of linear distance.

2) The proportions of missing taxa are quite high for such a biologically well-studied region. For example a total of 102 amphibian species is given, but only 59 are in the analysis. For squamates only 94 of 236 are included. With such large numbers missing, the results and conclusions of the study must be strongly affected by the sampling of species. Any non-random factors in the sampling of species (by family, by region, by range size etc) could have a major effect on the reported results.

3) The meaning of the correlation of phylogenetic beta diversity to relative phylogenetic diversity (PDrel) also needs consideration. PDrel is the result of a regression, reflecting both tree shape and the phylogenetic dispersion (or clustering) of the local species. Phylo beta also reflects these two factors of the local assemblages, as well as their degree of overlap between cells. So what does it mean that one metric of diversity on the tree explains another, based on the same tree and occurrences? I am not saying that this correlation should not be used, but they clearly cannot be independent of each other given how they are calculated. So the authors should clearly explain what a correlation (or deviation) between phylo beta and PDrel would mean, to justify use of the metric, before delving into the specific results for the taxa studied.

Also, high PDrel need not be due to old lineages (line 404), right? Co-occurring taxa from distantly related clades should be sufficient. And similarly, why must low PDrel require isolated radiations of a few closely related lineages? The PD part of the metric should not 'care' whether the species are

isolated (endemic).

4) The novelty of the approach - in isolating the environmental drivers of phylo beta diversity, and in looking at this relationship through evolutionary time - is strongly stated in the introduction. But later on a number of other studies are mentioned which also do parts of this, such as the analyses of geographic v environmental distance cited at lines 56-9. And then, that 'Penone et al. were the first to remove the effect of spatial distance on phylo- β in their global analysis' (lines 217-8).

This discussion of precedents and related studies is thorough and well presented. But the claims of novelty in the current study need to be more carefully explained or qualified, in particular that 'environmental patterns of phylo- β have not yet been analysed in a purely environmental context where the effect of geographic space is removed' (line 47).

*** Specific suggestions ***

Figure 1 and its caption are complex. If possible, reword the caption to be simpler to interpret. And on the right side of the figure labels strata 1 .. strata 4 should be 'stratum 1..' as each label refers to just one stratum.

L94 (and throughout the manuscript). Taxonomic Diversity often means a metric like PD but measured on a taxonomic tree comprising multiple taxonomic levels. For example Clarke & Warwick 1998: "taxonomic diversity can be thought of as the average taxonomic 'distance' between any two organisms, chosen at random from the sample: this distance can be visualized simply as the length of the path connecting these two organisms, traced through (say) a Linnean [or phylogenetic] classification of the full set of species involved.

Could you instead change all references from 'taxonomic diversity' to another term such as 'species diversity' or 'species richness' ?

L97 Why should high TD (species richness) lead to high turnover? Wouldn't sites with with many species tend (in general) to span more of the tree, and thus share more branch length (if not more species) with other sites, while less speciose sites would (on average) be better able to occupy distinct parts of the phylogeny.

Figure 1, L95, 101. The early mentions of strata left me confused, until I read the methods. Perhaps a line or two explaining the way the term strata (& stratum) is used, before they first appear in figure 1, would be helpful.

L135-9 The results here seem important enough to be shown in the main paper. But they depend on the difference between maps in Figure 3 and supplementary. Is it possible to show both endotherms and ectotherms in Fig 3 to bring out the comparison which is discussed (maybe at a similar size to Fig. 2).

L163 "Relative phylogenetic diversity for squamates was high across all environmental conditions". What does high across all conditions mean, given that PDrel is a residual and thus should be balanced between positive and negative values?

L163 please reword 'got low'. Perhaps 'became low' ?

L221-3 This was also shown by Rosauer et al 2014

L238-43 Nice result, well explained.

L301-2 How do exclusions in Easterns Europe relate to Fig. 1?

L422 "We used the rasterized version". Isn't this covered at line 416?

*** Minor wording etc ***

L22-23 The first sentence of the abstract could benefit from rewording. 'lineage distance' needs some unpacking or explanation, so a more standard term might be clearer (perhaps phylogenetic or evolutionary distance). And 'instrumental' is probably not the intended meaning here. Crucial? Central?

L50-51 "the drivers behind the mechanisms". What are drivers behind mechanisms? Would either drivers or mechanisms mean the same thing here?

L61 'is' should be 'are'

L128 'clear barrier' is probably not the right expression here. It sounds like a physical barrier in biogeography. How about a 'clear difference' or 'clear distinction'.

L187 'scales' why plural? How about 'scale'?

L204 'monotonously' probably not the word intended - usually means without variation in the sense of boring.

L249 'Generally, this likely'. Please reword.

L261 and 269 "the found transition", "The found patterns". Please change word order - eg "the transition zones found ..."

L286 "along environmental space" Along usually refers to something in one dimension. So maybe 'across environmental space' or 'along environmental gradients'...

L350-52 Please reword this sentence for clarity. Perhaps as two sentences.

L366 should be "number of strata"

L449 "that do not correlate overly high" Please reword for clarity and grammar.

Reviewer #2:

Remarks to the Author:

General Comments

This is a very nice paper dealing with phylobeta diversity in Eupore, using a new approach to decouple environmental and geographic distance to better understand processes underlying niche evolution. The paper is well written, with up-to-date analyses based on sounding data, and discussion/conclusions are firmly grounded on these empirical results.

In general, I would only caution for the use of word "niche size" (e.g., lines 66, 81, 251, 407, actually throughout the text), and better define this in advance. In a general sense, as pointed out in line 81, this is ok, but when you move to a more "operational" approach (e.g., Fig. 1), this may be a bit more complicated and more explanations may be necessary, in terms of the relationship between fundamental, existing and realized niches (I suggest looking at the recent paper by Soberon & Arroyo-Pena 2017 published in PLoS (<https://doi.org/10.1371/journal.pone.0175138>)). I agree that at this point it is not possible to empirically deal with these differences between niche "components" or "fractions", and I trust that at macroecological scales your results based on realized ranges are robust (despite the lack of correlation between physiological experiments and realized niches based on geographic ranges). However, perhaps it would be nice to give a short message on this.

Specific and Minor comments:

Line 70-71 – I'm not sure that most ecologists will follow you on pleiotropic effects, perhaps better explain this (this may be a quite complex issue, and you are assuming that niche dimensions are under strong genetic control?);

Line 73-74 – Something seems to be missing from this phrase?

Line 104 – about your goals, ok, but 1) is quite well known for beta-diversity in general (which is correlated with phylobeta), and there is a lot of literature on "distance decay" in exponential form (perhaps this should be mentioned here, and perhaps better tested here?)

Line 115 – this is because niche evolution is assumed to be different from a neutral/Brownian model, and also because there are phylogenetic non-stationarity and scale effects (as pointed out by one of the authors in recent papers);

Line 185 – Well, I see your point, but this is really not a novel "method" (it is only a partial regression approach – see below);

Line 344 – This is an important issue. Mathematically it is not hard to see why you used this delta model, but I'm afraid a better justification is needed here. One of the main points of people defending the "model-based" approaches in comparative analyses is that they can understand some meaning behind this, and I don't see how this will match any processes related to niche evolution. One can empirically compare alternative models and see how niches evolve in a given lineage, and there are indeed many papers testing this (and you could run this to your own data). Transform the branches according to a given model assume that this model fits data well, in my opinion, and must make sense in a biological context. You are using this empirically to give more weight to recent or deep branch, but if this is actually the goal, I would define this more explicitly and get the pairwise phylogenetic distances and explicitly truncate them at some distance in time, producing phylogenetic connections (see papers by Legendre in a spatial context, and many papers on correlograms).

In a more general sense, although this is not well advanced for phylogenetic analyses, I suggest you look at many papers on variance partition and analyses on beta diversity, especially those from Pierre Legendre, Pedro Peres-Neto and Stefan Dray, among others. One main discussion that you may take into account, defended by these authors, is that perhaps using eigenvectors from such matrices is more efficient than matrix comparison (but see Tuomisto's replies).

386-387 – Ok, this is a partial regression, not big deal (that's why I think that this is really not a novel

method – although I agree that it is nice to warn people about partitioning the effects of geography and environment on phylobeta). Many people may complain about using the residuals and actually getting only the “a” component (in Legendre’s jargon). The only real issue with this, as you point out, is that results will depend on the amount of the “b” (overlap”) component. I’m not sure if your results will really hold with high overlap components between geography and environment, and I’m afraid that overlaps are in general high (although I’m not sure for your particular case with phylobeta). Perhaps explicitly doing a partial regression and getting the a,b, and c, components would be nice.

419 – You said in the beginning that your phylobeta refers to turnover only component, so I’m not sure why using this “relative phylogenetic diversity” in this model.

POINT-BY-POINT RESPONSE LETTER

Please find our point-by-point response to comments below. They are displayed in *italicized Arial font in blue* and examples of changed manuscript text are pasted in "Times New Roman" font. The uploaded revised manuscript has **all changes marked in red**.

REVIEWERS REPORTS

Reviewer #1:

This manuscript examines the environmental correlates of phylogenetic beta diversity for 4 major tetrapod radiations in Europe. The key innovation of the study is to control for the component of beta diversity which is explained by geographic distance in order to reveal the component of compositional turnover which is specific to differences in environment.

The paper is well written, with thorough and interesting analysis. The rationale and findings of the study are well argued.

There are several concerns which it would be important to address. None of these are fatal, but they may require some reanalysis or changes to framing of the problem and results.

BS et al.: We highly appreciate the extensive and insightful comments by reviewer #1. We did our best to implement all concerns and suggestions of reviewer #1 and we are convinced that they led to more robust results and a clearly improved manuscript. More specifically, we reanalyzed all main and supplementary analyses and updated all graphs and figures in order to correct phylobeta for land distance, not for geographic (Euclidean) distance. Although it had no effect on the results and the main findings, we found it an excellent idea to correct for "land distance", as it makes theoretically much more sense. Not doing so might raise exactly the questions rev. #1 has raised. The results of a variant of "land distance" calculation and the previous version of geographic distance correction are included as robustness analyses in the supplementary analyses of the new version. We also included significant changes with regards to the other three main points. This led to a better presentation of the novelty of the study, and clarifications about the use of macroecological characteristics for interpreting phylobeta patterns.

1) The method used, evaluating phylogenetic turnover (phylo beta) in environmental rather than geographic space, is an effective way to remove the effects of geographic distance. But can the resulting phylo beta patterns can be interpreted as representing correlates of environment alone? Geographic distance is important to beta diversity as a surrogate for connectivity or isolation between communities. But connectivity depends on what lies between two locations, not just on the distance. For example, Italy and the Balkans are geographically close, but separated by the Adriatic Sea, while a similar distance in another part of Europe may have few significant barriers. And of course there are the Alps as a barrier (and corridor). So, the patterns which the ms presents as purely about environment, could be expected to actually reflect both environment, and connectivity.

For example could this help explain the higher turnover found in warmer environments (Fig 2 - amphibians, squamates). This is presented in terms of temperature, but could relate to sea barriers between major land areas of southern Europe, which contribute to higher phylogenetic turnover per unit of distance. The problem is complex, but the authors need to be more nuanced about the study's ability to "evaluate phylo beta in a purely environmental context" (lines 49-50). One option to consider would be to use land distance between cells in place of linear

distance.

BS et al.: The problem is indeed complex, and the reviewer has raised an important point. As suggested, we have re-run all our analyses using “land distance” instead of “linear distance”. We did so by implementing a cost-path distance analysis and used two different penalties for trajectory through sea water (10x/100x) compared to trajectories over land. As also big islands (e.g. Ireland, England, etc.) are included in our study, it does not make sense to increase the penalty overly high, because the calculated distance would tremendously increase for pixels separated by sea. On the other hand, a too low penalty would not make the path track land instead of water when e.g. comparing the pixels separated by the Adriatic Sea.

Our analyses showed that both the 10x and the 100x penalty for crossing over sea did not change the findings of our analyses. The patterns found along environmental gradients remained the same. We nevertheless decided to change all results in the manuscript according to the suggestions by reviewer #1. We calculate “land distance” by implementing a least-cost path distance with a 10-fold penalty for sea trajectories. The results for the “linear distance” and for the 100-fold penalty for sea trajectories are included in the supplementary material as robustness analyses (see new supplementary Fig. 12). We adjusted all figures (in the main manuscript and the supplementaries) using the land distance approach and we adjusted the methods section accordingly.

A correction for land barriers is more complex and theoretically less straightforward or meaningful in our context. The set of species we analyze here is not a priori restricted to specific habitats (e.g. warm-temperate environments), but instead varies widely in environmental preferences, as these species inhabit all the different habitats in Europe. Two species that can be found in one pixel thus can have very different ecological niches, and it is thus not straightforward to assume that two pixels separated by a mountain would be a barrier for both and requiring them to both move around the mountain to track suitable habitats. And since we often have many species in pixels we compare, the problem becomes both somewhat intractable and unrealistic. The only meaningful way to track the problem is likely to assume that all species that inhabit two neighboring environmental domains would prefer to migrate in this narrow domain and increasingly avoid other domains that increasingly differ ecologically from these specific environments. This is not a very realistic assumption. In fact, each of the species in such an assemblage may differ quite substantially in its ecological requirements, and the conditions in these neighboring environmental domains simply represents the (climatic) niche overlap. Assuming preferential migration in the above mentioned narrow ecological domain would therefore assume 1) climatic niches to be much narrower than they realistically are, and 2) that all species are much more similar in their environmental preferences than they realistically are. On the contrary, the problem of land distance (as implemented above) is thus much different from implementing land barriers into a distance analysis, as all species analyzed are here land – not marine – species.

We therefore abstained from calculating “land barriers” for correcting geographical distance on turnover. Rather (as mentioned above), we implemented the “land distance” approach and re-calculated all follow up analyses in the main results and in the supplementary materials. We replaced “geographic distance” by “geographic land distance” throughout the manuscript, when referring to our analyses.

Introduction

L107 ff: “First, we map phylo- β across environmental space after having removed the effect of geographic land distance among sample points on phylo- β (see Fig. 1 and Methods)...”

Methods

L406 ff: “Geographic distance is calculated here as “land distance”, thus penalizing trajectories through sea by a factor of 10 (using a least-cost path distance analysis with 10x higher costs over sea compared to land). Robustness analyses using both an even higher sea trajectory penalty (100x) and using “linear” (Euclidean) distance with no penalty for sea trajectories are given in Supplementary Fig. 12.”

2) The proportions of missing taxa are quite high for such a biologically well-studied region. For example a total of 102 amphibian species is given, but only 59 are in the analysis. For squamates only 94 of 236 are included. With such large numbers missing, the results and conclusions of the study must be strongly affected by the sampling of species. Any non-random factors in the sampling of species (by family, by region, by range size etc) could have a major effect on the reported results.

BS et al.: We realized that we did not explain this correctly, which has led to misunderstandings. We indeed included almost all species that are actually present in the study area (only very few are missing due to mismatch between the phylogeny and the distribution data). The phylogeny of Roquet et al. was actually built for the set of species included in the distribution data. So, the difference between the old version of the manuscript (e.g. total of 102 amphibians etc.) to the correct numbers (total of 67 amphibians, etc.; see below) comes from the spatial coverage of the data sources. The previous numbers referred to the original spatial extent of the distribution data, which included a broader range of Europe. We excluded smaller and less connected Islands and some parts of Eastern Europe (see methods L310ff). The table below gives a quick overview of the number of species included. We revised the section in the methods accordingly.

Species group	Total in Study area	Roquet Phylogeny	Hedge Phylogeny smoothed / unsmoothed
Amphibians	67	59	62
Squamates	95	94	83
Mammals	161	157	158 / 153
Birds	414	384	387 / 376

Methods

L321 ff: “Within our study area, a total of 67 amphibians, 95 squamates, 161 mammals, and 414 birds were present”

3) The meaning of the correlation of phylogenetic beta diversity to relative phylogenetic diversity (PDrel) also needs consideration. PDrel is the result of a regression, reflecting both tree shape and the phylogenetic dispersion (or clustering) of the local species. Phylo beta also reflects these two factors of the local assemblages, as well as their degree of overlap between cells. So what does it mean that one metric of diversity on the tree explains another, based on the same tree and occurrences? I am not saying that this correlation should not be used, but they clearly cannot be independent of each other given how they are calculated. So the authors should clearly explain what a correlation (or deviation) between phylo beta

and PDrel would mean, to justify use of the metric, before delving into the specific results for the taxa studied.

BS et al.: Phylogenetic beta diversity (phylo- β) as used here reflects the “true turnover”, and thus is not affected by differences in phylogenetic diversity between two sites. Variation in phylo- β between environmental domains aligned along environmental gradients can thus originate from the following three effects (and these effects might have different weights in the different species groups): (1) species inhabiting specific environmental domains can have small or large environmental niches, thus changes along env. gradients may result in higher or lower phylo- β ; and/or (2) environmental domains can contain high or low species richness (SR), thus changes along env. gradients can affect variations in phylo- β , and the sign (increase or decrease in phylo- β with changes in species richness) depends on the degree of niche conservatism (less specious sites either tend towards high or low phylo- β depending on the degree of niche conservatism, while more specious sites have a tendency towards average phylo- β as a result of larger amounts of shared branch lengths); and/or (3) species inhabiting specific environmental domains can be phylogenetically relatively distantly (high PDrel) or closely (low PDrel) related and thus changes along the env. axis may result in higher or lower phylo- β .

We thus hypothesize that the three factors can jointly (“and” relation) or independently (“or” relation) explain variations in phylo- β . Therefore, PDrel is not the only factor that can drive phylo- β , as variations in NS or SR can also directly affect phylo- β . With the analysis in Figure 4, we aim to assess how these factors combine to explain phylo- β along environmental gradients. It is obvious that in mammals and birds, the predictive power is lower, as we have less structure in phylo- β along environmental gradients. But we also find strong differences between the two ectotherm groups. Phylo- β of amphibians is primarily explained by variations in SR and PDrel (which are uncorrelated) and not by NS (which is always comparably low in this clade). Phylo- β of squamates is primarily explained by SR and NS but not by PDrel, which shows variation but is not correlated with phylo- β . This demonstrates that phylo- β can be mis-interpreted as being driven by PDrel (differences in closely vs. distantly related species) alone. If the latter would be the only driver, then indeed relating PDrel to phylo- β would not make much sense (as reviewer #1 argues). But because phylo- β and species richness are comparably highly correlated, it is not necessarily variation in PDrel that drives variation in phylo- β . The two ectotherms thus differ fundamentally in the way they adapted evolutionarily to the regions where low phylo- β occurs (cold/wet). In amphibians, only very few and closely related taxa adapted to these environments (PDrel is low, here). In squamates, also few, but less closely related taxa have adapted to these colder/wetter environments (and PDrel is comparably high across the whole environmental domain).

To improve the manuscript, we made several changes to the introduction and to the discussion. The introduction now better reflects the expectations behind analyzing macroecological characteristics for explaining phylo- β patterns. We also adjusted the Figure 1 accordingly.

Introduction

L86 ff: “Regions of high relative phylogenetic diversity are thought to have experienced high diversification rates of multiple lineages or immigration of multiple lineages that radiated successfully. In contrast, regions of low relative phylogenetic diversity may be associated with

large radiations of few, closely-related lineages indicating that other lineages from the same taxon have not successfully colonized¹⁸. Therefore, trends in relative phylogenetic diversity along environmental gradients may strongly affect the emergence of phylo-β patterns and point to the degree of niche conservatism.”

L98 ff: “Therefore, changing niche sizes along an environmental gradient may strongly affect the emergence of phylo-β patterns.”

L99 ff: “In summary, we hypothesize that these three macroecological characteristics (species richness, relative phylogenetic diversity and niche size) can jointly or independently explain variations in phylo-β (Fig. 1c).”

Discussion

L277 ff: “Compared to amphibians, the phylogenetic structure was less pronounced in squamates, meaning that the few clades that have adapted to cold-wet domains are less closely related than those in amphibians (e.g. indicated by the low importance of the relative phylogenetic diversity in explaining phylo-β patterns).”

Also, high PDrel need not be due to old lineages (line 404), right? Co-occurring taxa from distantly related clades should be sufficient. And similarly, why must low PDrel require isolated radiations of a few closely related lineages? The PD part of the metric should not 'care' whether the species are isolated (endemic).

BS et al.: Indeed, that’s correct. We used a wrong wording here (“old” instead of “distantly related”) and “isolated” was also used inappropriately. Thank you for your careful reading and pointing this out. We changed it accordingly in the mentioned part in the methods, but also in the introduction.

Introduction

L86 ff: “Regions of high relative phylogenetic diversity are thought to have experienced high diversification rates of multiple lineages or immigration of multiple lineages that radiated successfully. In contrast, regions of low relative phylogenetic diversity may be associated with large radiations of few, closely-related lineages indicating that other lineages from the same taxon have not successfully colonized¹⁸.”

Method

L430 ff: “That is, high relative phylogenetic diversity highlights areas of co-occurring taxa from distantly related lineages or it represents species rich areas due to high immigration rates of multiple lineages¹⁸. In contrast, low relative phylogenetic diversity relates to radiations of few closely related lineages¹⁸.”

4) The novelty of the approach - in isolating the environmental drivers of phylo beta diversity, and in looking at this relationship through evolutionary time - is strongly stated in the introduction. But later on a number of other studies are mentioned which also do parts of this, such as the analyses of geographic v environmental distance cited at lines 56-9. And then, that 'Penone et al. were the first to remove the effect of spatial distance on phylo-β; in their global analysis' (lines 217-8).

This discussion of precedents and related studies is thorough and well presented. But the claims of novelty in the current study need to be more carefully explained or qualified, in particular that 'environmental patterns of phylo-β; have not yet

been analysed in a purely environmental context where the effect of geographic space is removed' (line 47).

BS et al.: Thank you, this is important. We have restructured the whole introduction and more carefully explain how far other studies have gone. In addition, we explain the novelty of our study more specifically. By doing so, we deleted the sentence in line 47 (of the old manuscript) "environmental patterns of phylobeta have not yet...", because the statement was misplaced and unclear. We also changed Figure 1 accordingly.

Introduction

L103 ff: "Here, we quantify and decipher for the first time the patterns of phylo- β in environmental space for European tetrapods. We do so by analyzing phylo- β patterns across European samples of tetrapod distributions, and by mapping phylo- β in equidistant steps (environmental strata, hereafter) along environmental gradients (Fig. 1). The novelty of our approach is two-fold. First, we map phylo- β across environmental space after having removed the effect of geographic land distance among sample points on phylo- β (see Fig. 1 and Methods), rather than testing distance decays or to what degree phylo- β can be partitioned into geographic and environmental distance effects. Second, we seek to explain the resulting phylo- β patterns by means of macroevolutionary characteristics to assess the different processes shaping phylo- β ."

Discussion

L228 ff: "Rather, it is the positioning within an explicit environmental framework that explains high vs. low phylo- β values, as was hypothesized earlier by Warren et al.²⁰ from a conceptual perspective and supported by the relationship found between phylogenetic turnover and variation in local environmental conditions by Rosauer et al.¹³. However, phylo-GDM calculates turnover among all sample points. Therefore, the phylo- β patterns found are not explicitly analyzed for equidistant progression along environmental gradients as in our study and may thus tend to average phylo- β – environment relationship over various environmental extents.

***** Specific suggestions *****

Figure 1 and its caption are complex. If possible, reword the caption to be simpler to interpret. And on the right side of the figure labels strata 1 .. strata 4 should be 'stratum 1..' as each label refers to just one stratum.

BS et al.: We acknowledge that this was not clear enough and we rephrased the caption. We corrected "strata" to "stratum", thanks for spotting this. We changed the Figure 1 to account for comments by both reviewers, and the changed figure now presents the novelty of our approach more clearly.

Introduction

*L59 ff: "**Fig. 1.** Analysis and presentation of phylo- β structure across environmental space. **a** Relationship between phylo- β (corrected for geographical land distance, i.e. residuals of phylo- β ~ land distance) and environmental distance among pairs of sample points reveals large variation, indicating that distances are not necessarily good predictors of phylo- β . Example points refer to panel **b**. **b** Illustration of a hypothesized structuring of phylo- β between neighboring equidistant strata (dotted and solid quadrats) resulting in environmental zones of low (domain A) or high (domain B) values. The mapped phylo- β values represent the means of all pairwise phylo- β values (black dots and black crosses in **a**) of sample sites*

between neighboring strata (red dots in neighboring strata within domain A, red crosses in neighboring strata within domain B). The sample sites between neighboring strata share a relatively short environmental distance (see panel a for illustration), but originally had a large variation in spatial distance (for which it had been corrected for). c Illustration of possible macroecological drivers on phylo- β among sample points (e.g. sites 1, 2 for domain B; sites 3, 4 for domain A). Phylo- β is hypothesized to be explained jointly (“and” relation) or independently (“or” relation) by niche size (NS), species richness (SR), relative phylogenetic diversity (PDrel).”

L94 (and throughout the manuscript). Taxonomic Diversity often means a metric like PD but measured on a taxonomic tree comprising multiple taxonomic levels. For example Clarke & Warwick 1998: "taxonomic diversity can be thought of as the average taxonomic 'distance' between any two organisms, chosen at random from the sample: this distance can be visualized simply as the length of the path connecting these two organisms, traced through (say) a Linnean [or phylogenetic] classification of the full set of species involved.

Could you instead change all references from 'taxonomic diversity' to another term such as 'species diversity' or 'species richness' ?

BS et al.: We thank the reviewer #1 for raising this point. We agree and changed 'taxonomic diversity' to 'species richness' throughout the manuscript.

L97 Why should high TD (species richness) lead to high turnover? Wouldn't sites with many species tend (in general) to span more of the tree, and thus share more branch length (if not more species) with other sites, while less speciose sites would (on average) be better able to occupy distinct parts of the phylogeny.

BS et al.: Please see our response to the main pt. 3 above by reviewer #1. We argue that changes in species richness along env. gradients can affect variations in phylo- β , and the sign (increase or decrease in phylo- β with changes in species richness) depends on the degree of niche conservatism because less speciose sites either tend towards high or low phylo- β depending on the degree of niche conservatism, while more speciose sites have a tendency towards average phylo- β as a result of larger amounts of shared branch lengths (as rev. #1 points out). Therefore, the phylo- β trend towards more speciose sites can either be positive or negative, depending on the degree of niche conservatism. As outlined under pt. 3 above, we have adjusted the manuscript accordingly.

Figure 1, L95, 101. The early mentions of strata left me confused, until I read the methods. Perhaps a line or two explaining the way the term strata (& stratum) is used, before they first appear in figure 1, would be helpful.

BS et al.: We followed this suggestion and now provide a short explanation of the term strata/stratum.

Introduction

L104 ff: " We do so by analyzing phylo- β patterns across European samples of tetrapod distributions, and by mapping phylo- β in equidistant steps (environmental strata, hereafter) along environmental gradients (Fig. 1).”

L135-9 The results here seem important enough to be shown in the main paper. But they depend on the difference between maps in Figure 3 and supplementary. Is it possible to show both endotherms and ectotherms in Fig 3 to bring out the comparison which is discussed (maybe at a similar size to Fig. 2)."

BS et al.: Thank you for the suggestion. We changed Figure 3 accordingly and now show both the results of the endo – and ectotherms.

L163 "Relative phylogenetic diversity for squamates was high across all environmental conditions". What does high across all conditions mean, given that PDrel is a residual and thus should be balanced between positive and negative values?

BS et al.: Thank you, we were not precise enough in our wording. We now changed the wording to "positive" and "negative" in this and also in the previous sentence (instead of using the terms "high" and "low"), and in the discussion. In the results part, the manuscript now reads:

Results

L169 ff: "Relative phylogenetic diversity was clearly structured in amphibians, mammals and birds, with a tendency towards negative relative phylogenetic diversity values in cold-wet domains. Relative phylogenetic diversity for squamates was negative in comparably hot and comparably cold conditions and positive in-between, and overall positive in very warm conditions."

L163 please reword 'got low'. Perhaps 'became low' ?

BS et al.: We changed it to "was negative" (L171), also in response to the comment above.

L221-3 This was also shown by Rosauer et al 2014

BS et al.: We added this reference accordingly, thank you for calling attention to it:

Discussion

L228 ff: "Rather, it is the positioning within an explicit environmental framework that explains high vs. low phylo- β values, as was hypothesized earlier by Warren et al.²⁰ from a conceptual perspective and supported by the relationship found between phylogenetic turnover and variation in local environmental conditions by Rosauer et al.¹³. However, phylo-GDM calculates turnover among all sample points. Therefore, the phylo- β patterns found are not explicitly analyzed for equidistant progression along environmental gradients as in our study and may thus tend to average phylo- β – environment relationship over various environmental extents."

L238-43 Nice result, well explained.

BS et al.: Thank you for the positive feedback.

L301-2 How do exclusions in Easterns Europe relate to Fig. 1?

BS et al.: Thanks for spotting this. It should be "Supplementary Figure 1a" in which

the study area is represented (corrected now on L315).

L422 "We used the rasterized version". Isn't this covered at line 416?

BS et al.: Indeed, it's a repetition. We therefore deleted this sentence.

*** Minor wording etc ***

L22-23 The first sentence of the abstract could benefit from rewording. 'lineage distance' needs some unpacking or explanation, so a more standard term might be clearer (perhaps phylogenetic or evolutionary distance). And 'instrumental' is probably not the intended meaning here. Crucial? Central?

BS et al.: Thank you for your suggestions to re-word the first sentence. We changed it accordingly:

Introduction

L22 ff: "Phylogenetic turnover quantifies the evolutionary distance among species assemblages and is central to understand the main drivers shaping biodiversity."

L50-51 "the drivers behind the mechanisms". What are drivers behind mechanisms? Would either drivers or mechanisms mean the same thing here?

BS et al.: Thank you for pointing this out. Yes, they meant the same thing. We realized that the content of this sentence was already included in a previous sentence and decided to remove it. The previous sentence including this content is:

Introduction

L40 ff: "Phylogenetic turnover (phylo- β), which quantifies the phylogenetic distance among communities (i.e. pair-wise compositional turnover), can be used to explore the main drivers behind biodiversity patterns^{5,6} and to define⁷ or explain¹ the structuring of biodiversity in geographical space."

L61 'is' should be 'are'

BS et al.: Thanks for spotting this. This sentence was deleted in the process of rephrasing the introduction following previous suggestions of reviewer #1.

L128 'clear barrier' is probably not the right expression here. It sounds like a physical barrier in biogeography. How about a 'clear difference' or 'clear distinction'.

BS et al.: We changed it to "clear difference" (L135).

L187 'scales' why plural? How about 'scale'?

BS et al.: We changed it to "scale" (L194). Thanks for spotting this.

L204 'monotonously' probably not the word intended - usually means without variation in the sense of boring.

BS et al.: This is correct, we changed the meaning to "not strongly correlated with..."

(L212)

L249 'Generally, this likely'. Please reword.

BS et al.: We deleted "Generally" (L261).

L261 and 269 "the found transition", "The found patterns". Please change word order - eg "the transition zones found ..."

BS et al.: Done (L273/282).

L286 "along environmental space" Along usually refers to something in one dimension. So maybe 'across environmental space' or 'along environmental gradients'...

BS et al.: We changed it to "across environmental space" (L299).

L350-52 Please reword this sentence for clarity. Perhaps as two sentences.

BS et al.: We split this sentence in two. It now reads:

Methods

L368 ff: "We chose two environmental variables that represent important climate constraints on species distributions, niche evolution, and adaptations. Specifically, they reflect adaptations to extreme cold or extreme drought."

L366 should be "number of strata"

BS et al.: Changed it (L384).

L449 "that do not correlate overly high" Please reword for clarity and grammar.

BS et al.: I changed it to "...that show low correlations across Europe." (L476)

Reviewer #2:

General Comments

This is a very nice paper dealing with phylobeta diversity in Eupore, using a new approach to decouple environmental and geographic distance to better understand processes underlying niche evolution. The paper is well written, with up-to-date analyses based on sounding data, and discussion/conclusions are firmly grounded on these empirical results.

BS et al.: Thank you for this positive feedback. We highly appreciate your careful and insightful review and we did our best to implement your suggestions. We are very happy about the raised points and think they helped to improve the manuscript significantly. Specifically, the points raised about niche size and relative phylogenetic

diversity have stimulated our thinking and led to several clarifications. We further improved the declaration of novelty in this paper. To do so, we now abstained from referring to a novel method (which it is not, as rev. #2 correctly pointed out), and better circumscribe the elements of novelty and the specific goals, expectations, and the general idea of the study.

In general, I would only caution for the use of word “niche size” (e.g., lines 66, 81, 251, 407, actually throughout the text), and better define this in advance. In a general sense, as pointed out in line 81, this is ok, but when you move to a more “operational” approach (e.g., Fig. 1), this may be a bit more complicate and more explanations may be necessary, in terms of the relationship between fundamental, existing and realized niches (I suggest looking at the recent paper by Soberon & Arroyo-Pena 2017 published in PLoS (<https://doi.org/10.1371/journal.pone.0175138>)). I agree that at this point it is not possible to empirically deal with these differences between niche “components” or “fractions”, and I trust that at macroecological scales your results based on realized ranges are robust (despite the lack of correlation between physiological experiments and realized niches based on geographic ranges). However, perhaps it would be nice to give a short message on this.

BS et al.: Thank you, this is a fair point. We now refer to the Soberon and Arroyo-Pena (2017) paper, which excellently outlines the issue. We also state that evolution primarily affects the fundamental niche size as suggested in this paper and in the comment by reviewer #2. However, we also add the notion that a species can improve its competitive capacity (e.g. through improved resource use capacity), which can directly affect the competitive balance and thus the realized niche (without necessarily affecting the fundamental niche). We also state that fundamental niche adaptations of other species may affect the realized niche of a target species. And we finally state that throughout the paper we now refer to the realized niche size when using the term NS.

Introduction

L92 ff: “Evolutionary and ecological processes also affect the realized niche size of species. Evolution primarily affects the fundamental niche of species¹⁹. Yet, evolutionary adaptations in the competitive performance of a target species (e.g. through improved resource use efficiency) may also affect the realized niche of that species, and so does the adaptation of the fundamental niche of other, interacting species. Hereafter, niche size thus refers to the size of the realized niche available in a given region.”

Method

L433 ff: “The size of the realized niche was calculated ...”.

Specific and Minor comments:

Line 70-71 – I’m not sure that most ecologists will follow you on pleiotropic effects, perhaps better explain this (this may be a quite complex issue, and you are assuming that niche dimensions are under strong genetic control?);

BS et al.: We explicitly state now, that specialization can be accompanied by a cost (e.g. pleiotropic effects). We changed the sentence to:

Introduction

L76 ff: “For example, the amplitude of adaptive radiations may be constrained in extreme environmental domains (distinct regions within environmental space) due to accompanied costs of niche specialization (predicted by e.g. the theory of antagonistic pleiotropy or mutation accumulation)^{16,17}. In contrast, less extreme environmental domains might promote higher diversification due to reduced costs of niche specialization that allow for easier radiations into neighboring habitats¹⁶.”

Line 73-74 – Something seems to be missing from this phrase?

BS et al.: I rephrased to:

Introduction

L81 ff: “Therefore, different environmental domains may contain low or high species richness or phylogenetic diversity.”

Line 104 – about your goals, ok, but 1) is quite well known for beta-diversity in general (which is correlated with phylobeta), and there is a lot of literature on “distance decay” in exponential form (perhaps this should be mentioned here, and perhaps better tested here?)

BS et al.: We would like to ascertain that we are not interested in distance decay. The reviewer #2 is correct, there is a lot of literature on this topic. In distance decay analyses, one is interested to vary (geographic or environmental) distances and then to assess the shape of similarity-decay (in the compositional structure of plant or animal communities) with increasing distances. Usually, in such studies, a large variation in (phylo-)beta remains at a given distance, and even already over comparably short distances. This is the core difference (and what we think innovative part) of our paper. We now make this even clearer throughout the manuscript. Our intention is to demonstrate that when moving forwards along an environmental gradient with always the same distance (same difference in e.g. climatic parameters), phylobeta may a) show clear structuring, and b) allow for analyzing some of the evolutionary drivers behind. Neither beta nor phylobeta have been mapped in environmental space (while they have been mapped before in geographic space, e.g. Buckley & Jetz, 2008, PNAS for beta in global birds and amphibians, Peixoto et al., 2017, GEB for phylobeta in different global mammalian clades). We argue here, that specifically the mapping in environmental space along comparably short, equidistant steps, allows for analyzing important drivers behind (phylo-)beta.

In order to make this point clearer, we have adjusted the text in different locations. However, we believe that it is still a valid goal to analyze the phylobeta patterns in environmental space (goal 1). Phylobeta may or may not reveal structure, depending on how the evolutionary processes have affected the four major clades of European tetrapods. The following are examples of how we improved the text:

Introduction

L103 ff: “Here, we quantify and decipher for the first time the patterns of phylo- β in environmental space for European tetrapods. We do so by analyzing phylo- β patterns across European samples of tetrapod distributions, and by mapping phylo- β in equidistant steps (environmental strata, hereafter) along environmental gradients (Fig. 1). The novelty of our approach is two-fold. First, we map phylo- β across environmental space after having removed the effect of geographic land distance among sample points on phylo- β (see Fig. 1

and Methods), rather than testing distance decays or to what degree phylo- β can be partitioned into geographic and environmental distance effects. Second, we seek to explain the resulting phylo- β patterns by means of macroevolutionary characteristics to assess the different processes shaping phylo- β .“

Discussion

L228 ff: “Rather, it is the positioning within an explicit environmental framework that explains high vs. low phylo- β values, as was hypothesized earlier by Warren et al.²⁰ from a conceptual perspective and supported by the relationship found between phylogenetic turnover and variation in local environmental conditions by Rosauer et al.¹³. However, phylo-GDM calculates turnover among all sample points. Therefore, the phylo- β patterns found are not explicitly analyzed for equidistant progression along environmental gradients as in our study and may thus tend to average phylo- β – environment relationship over various environmental extents.

Line 115 – this is because niche evolution is assumed to be different from a neutral/Brownian model, and also because there are phylogenetic non-stationarity and scale effects (as pointed out by one of the authors in recent papers);

BS et al.: Thanks for complementing this. We assume that reviewer #2 refers to the recent paper by Rolland et al. (2018). We now include it by extending the rationale why tetrapods are well suited study clades for our analyses:

Introduction

L119 ff: “Analyses are carried out separately for the four major clades of European tetrapods (i.e. amphibians, birds, mammals and squamates). These clades are particularly well suited for testing our hypotheses because the two major groups (ecto- and endotherms) differ in their thermal physiology, niche evolutionary rates, climatic tolerances, and dispersal capacities²⁴, leading to distinct broad-scale ecological patterns.”

Line 185 – Well, I see your point, but this is really not a novel “method” (it is only a partial regression approach – see below);

BS et al.: We admit that this was not termed correctly. In several locations throughout the manuscript, we have abstained from the formulation of a “novel method”. Rather, we explain in two sentences what the novelty is (introduction) and we explain also, that this is the first time that phylo- β is mapped in ecological space (introduction).

Introduction

L103 ff: “Here, we quantify and decipher for the first time the patterns of phylo- β in environmental space for European tetrapods.”

L106 ff: “The novelty of our approach is two-fold. First, we map phylo- β across environmental space after having removed the effect of geographic land distance among sample points on phylo- β (see Fig. 1 and Methods), rather than testing distance decays or to what degree phylo- β can be partitioned into geographic and environmental distance effects. Second, we seek to explain the resulting phylo- β patterns by means of macroevolutionary characteristics to assess the different processes shaping phylo- β .”

Methods

L415 ff: “However, in phylo- β analyses of continental scale, geographic and environmental

distances usually only explain a small proportion and the unexplained fraction is usually very high^{8,12}. Yet, our approach allows for mapping both the environmentally structured and the unexplained fraction directly in to environmental space in order to interpret the results in a direct ecological context, and not simply in a distance context.”

Line 344 – This is an important issue. Mathematically it is not hard to see why you used this delta model, but I’m afraid a better justification is needed here. One of the main point of people defending the “model-based” approaches in comparative analyses is that they can understand some meaning behind this, and I don’t see how this will match any processes related to niche evolution. One can empirically compare alternative models and see how niches evolve in a given lineage, and there are indeed many papers testing this (and you could run this to your own data). Transform the branches according to a given model assume that this model fits data well, in my opinion, and must make sense in a biological context. You are using this empirically to give more weight to recent or deep branch, but if this is actually the goal, I would define this more explicit and get the pairwise phylogenetic distances and explicitly truncate them at some distance in time, producing phylogenetic connections (see papers by Legendre in a spatial context, and many papers on correlograms).

BS et al.: Yes, this is actually the goal: to give more weight to recent or deep branches, rather than comparing niche evolution in different lineages. We understand that the approach developed by Cavender-Bares & Reich (2012, Ecology) and e.g. used by Groussin et al. (2012, Nature Communications) and Mazel et al. (2017, Global Ecol Biogeogr) does what is suggested by reviewer #2. When truncating a phylogenetic tree as proposed therein, one is pruning younger branches and explicitly testing for patterns that may have emerged by the older part of the tree alone. This is essentially similar to delta transformations with delta values smaller than 1, although implemented more rigorously, by removing the subtle differences that a small delta still gives to different species that are – in turn - all collapsed into one OTU when truncating the tree. The alternative approach of snipping older branches away by assembling them to polytomies at different ages as was applied by Rosauer et al. (2014, Ecography). The intention here is to see if younger branches carry more environmental structure than even the current phylogeny, because (as Rosauer et al. phrase it) “older branches might confound the relationship with current environment”. This is essentially similar to delta transformations with delta values larger than 1, although implemented more strictly, by removing the subtle differences of older branches in the rescaled tree.

We implemented the suggested transformation by truncating the tree at older nodes similar to Cavender-Bares & Reich (2012) or Mazel et al. (2017). We found that the results are essentially the same as with delta transformations, the differences are very small. We now include this result from truncated trees as an additional measure of robustness of the analyses pathway in the supplementary material, but decided to leave the delta-transformed trees in the main manuscript. This decision was taken because: a) the results are the same, b) we were not intended to explicitly date certain events but rather only to compare between clades, and c) it allows with the same transformation to test for giving weight to rather younger and to rather older branches.

Results

L145 ff: “These results are similar to what is found when truncating the tree at certain depths in the phylogeny rather than δ -transforming the branches (See Supplementary Fig. 4).”

Methods

L362 ff: “An alternative method of this analysis step would be to truncate the phylogeny at certain phylogenetic depths, as developed⁵⁸ and implemented^{59,60} in recent studies. For reasons of comparison, we also implemented this method. We pruned the phylogenies at five specific time periods (depths) and collapsed all descendant leaves of each of the branches encountered by the pruning (see Supplementary Fig. 4). The geographical distribution of these branches is defined as the union of the distributions of their descending leaves.”

In a more general sense, although this is not well advanced for phylogenetic analyses, I suggest you looking at many papers on variance partition and analyses on beta diversity, especially those from Pierre Legendre, Pedro Peres-Neto and Stefan Dray, among others. One main discussion that you may take into account, defended by these authors, is that perhaps using eigenvectors from such matrices is more efficient than matrix comparison (but see Tuomisto's replies).

BS et al.: This is indeed a tricky issue and no consensus seems to have established. The main question remains, whether beta-diversity (or phylobeta-diversity) as a (univariate) metric can be analyzed further for understanding drivers that affect variation in this metric (through linear models), or whether the full community behind the metric should be used to partition the variance (ordination approach). The quite intense debate between Legendre and his colleagues/collaborators (including Dray, Laliberté, Peres-Neto, etc.) on the one hand and Tuomisto & Ruokolainen (2006, 2008) on the other hand did, in our view, not fully resolve the problem. Much debate arose over the issue whether a specific question is of level 2 or level 3 (according to Tuomisto & Ruokolainen 2006) and whether matrix regression is deflated for level 3 questions. Tuomisto & Ruokolainen (2006, 2008) argue that it is fine to partition distance matrices (e.g. phylobeta), if this distance is the target to be analyzed and the species behind the metric are not further targeted. Legendre et al. (2008) answered by saying that the matrix regression approach is not necessarily acceptable and that 4 tests should be applied first. Tuomisto & Ruokolainen (2008) answered that 3 of these 4 issues are already covered by the statistical literature, and that the fourth is not really an issue.

So who is right? As non-statisticians, it is hard to fully judge. A comparative paper using simulated community data to decompose spatial and environmental partitions in explaining beta-diversity (B. Gilbert & J.R. Bennett, 2010, J.Ecol.) – a level 3 question, as is ours - did not really find a method that was clearly superior (irrespective of matrix regression or eigenvector approaches), as all failed somehow to perfectly reconstruct the designed variance proportions. A similar test has not yet been carried out for phylogenetic beta-diversity, although we assume that such a test would come up with similar results. When it comes to analyzing the variance proportions that explain phylogenetic beta-diversity, then many well-cited studies we are aware of have used the approach implemented here (e.g. Eiserhardt et al. 2013, Scientific Reports; Penone et al. 2016, Proc. Roy. Soc. B; Saito et al. 2015, J. Biogeogr.; etc.). Admittedly, there are also studies that use RDA or CCA-based approaches (e.g. Arnan et al. 2015, PeerJ; Roncal et al. 2011 Biotropica; or Moura et al. 2017, J. Animal Ecol.). We argue here that both options are suitable methods, and since we are only interested in phylobeta as metric, eigenvector-based approaches

are not the only solution.

However, we also want to stress that we start from the assumption that the variation explained is not the real target behind our approach. So, we likely have overstated the “variation partitioning approach” in the previous version of the manuscript. There always remains a large amount in unexplained variance in data that spans across such large geographic and environmental distances. This is e.g. visible in Penone et al. (2016, see pasted figure to the right), in Eiserhardt et al. (2013, Fig. 2), or also in our new figure 1a. In none of these cases, the explained variance was very high, highest in Eiserhardt, who used a clade (neotropical palms) with comparably narrow ecological requirements. Therefore, neither geo.dist nor env.dist explain a large proportion. And once geo.dist effects are removed, env.dist only explains only a small fraction (small slope of red line and large variation around line) in Penone et al. (2016, right). The main point (of novelty) we want to make is that we are not “per se” interested in the effect of distance as such, but to map the residual phylobeta variation in ecological space (our new figure 1b). By this we learn how phylobeta is structured in environmental space. We do so by mapping phylobeta between neighboring environmental strata only.

Therefore, we map the encircled variation (blue shade, panel c on top of the Penone figure), which all have relatively small environmental distance, and originally had a large variation in spatial distances (for which we have corrected). Therefore, we map the (a+d) part in environmental space (or simply phylobeta which we have corrected for geo.dist). So: thank you for raising the point above. This point became also clearer to us now that we revised the manuscript and it resulted in a change to figure 1 in order to clarify this point. We now tone down the “variation partitioning” element in the description of the method.

Having said that, we decided to keep the results as we have implemented them originally, but we added some explanation to the text that also eigenvector-based approaches would be suitable to remove the spatial effects on phylobeta, and we toned down the “variation partitioning” element of this analysis step. Note that in response to rev. #1, we had to replace geographic distance by “land distance”, which is the distance along terrestrial pixels.

Methods.

L410 ff: “The residuals from the regression represent the phylo- β fraction that is independent of the land distance among sampled sites. This approach is similar as the one used in previous studies^{8,12}. It represents an element of variation partitioning⁶⁴ and removes the spatial distance effect and also some fraction of the environmental distance effect, as the two effects are not fully independent. However, in phylo- β analyses of continental scale, geographic and environmental distances usually only explain a small proportion and the unexplained fraction is usually very high^{8,12}. Yet, our approach allows for mapping both the environmentally structured and the unexplained fraction directly in to environmental space in order to interpret the results in a direct ecological context, and not simply in a distance

From Penone et al. (2016) representing figure 2, where phylobeta (top) and the residuals of a phylobeta-geo.dist (bottom) are plotted. We have added the blue shade for illustration.

context. An alternative approach for removing the effect of geographic distance would be to use eigenvector-based techniques⁶⁵.”

386-387 – Ok, this is a partial regression, not big deal (that’s why I think that this is really not a novel method – although I agree that it is nice to warn people about partitioning the effects of geography and environment on phylobeta). Many people may complain about using the residuals and actually getting only the “a” component (in Legendre’s jargon). The only real issue with this, as you point out, is that results will depend on the amount of the “b” (overlap”) component. I’m not sure if your results will really hold with high overlap components between geography and environment, and I’m afraid that overlaps are in general high (although I’m not sure for your particular case with phylobeta). Perhaps explicitly doing a partial regression and getting the a,b, and c, components would be nice.

BS et al.: We agree about the statement of novelty. It is not the method that we consider novel (we now have adjusted the text), but rather the approach of mapping phylobeta in environmental space after controlling for the effect of geographic (or land) distance. As stated in the point above, the proportion of explained variance by distances is not the real issue. We wanted to map phylobeta across environmental space. And because neighboring environmental conditions can be both in close proximity or far apart geographically, we needed to remove the effect of spatial distance on the analyzed phylobeta values. Neither geographic nor environmental distance usually explain a large proportion of phylobeta if analyzed across large spatial and environmental distances, and the majority of variation remains unexplained (see also arguments on the previous point). But we are interested in positioning this unexplained variation into an explicit environmental space, in order to analyze its structuring. This means that we are not simply interested in the “a” part, but actually in the “a” AND the “d” (unexplained variance) part (see new figure 1a,b). Therefore, the “b” part (joint effect by geographic and environmental distance) is not that important. The question is rather, whether the “d” component of a simple distance-relationship reveals structure if phylobeta of the same short environmental distance is mapped in ecological space. We find that it does. But we nevertheless needed to remove the distance effect, such that the found pattern does not represent a hidden effect of uneven distances among neighboring environmental strata.

419 – You said in the beginning that your phylobeta refers to turnover only component, so I’m not sure why using this “relative phylogenetic diversity” in this model.

BS et al.: This point was also raised by reviewer #1 in his/her general point 3. We give a detailed answer why we find that relative phylogenetic turnover is an important but not the sole criterion that helps to understand the variation of phylobeta in environmental space.

Reviewers' Comments:

Reviewer #1:

Remarks to the Author:

Thank you to the authors, for a very thorough and thoughtful response to issues raised in the first review of this ms. And for making the changes really easy to follow in the revised ms.

The terminology used and rationale for the methods are significantly revised, and the calculation of geographic distance was replaced with a land distance metric. The authors have also been very thorough in testing robustness of the results to choices at many stages in the methods.

The data used throughout the study - true phylogenetic turnover (phylo- β) corrected for geographic distance - are the residuals of a GLM model (logistic regression) for each of the four tetrapod classes. But neither the original phylo- β results nor the results of these models are given.

So can I request one more addition to the results, which would be to include in the main paper the model fit (presumably r^2) for each of the geographic models, and in supplementary a scatterplot of phylo- β for each group with the fitted line plotted. This would give the interested reader a much clearer understanding of the basis of the residual values on which the study is built.

** Other suggestions **

Figure 1:

* why not make the dots and crosses on panels a and b the same colour?

* the y-axis label says '(residuals)' but the colour legend does not. At first I thought the colours represented actual phylo- β (which would add information), rather than duplicating the information given by y-axis position in panel 1. So could you label the colour legend to be clear it is also the residual values?

* in panel c, are NS and niche width the same thing? And in the figure legend.

lines 109-10 - 'to what degree phylo- β can be partitioned into geographic and environmental distance effects.' This raises again the intriguing question - how much phylo- β actually was explained by geographic distance??

lines 211-13 - 'The discontinuities in phylo- β along environmental gradients are generally congruent with the view that phylo- β is not strongly correlated with environmental distances, a common pattern found in empirical studies'

This seems to be a key finding of the study. Where are the results for the correlation of phylo- β to the two environmental dimensions?

** Minor wording etc **

line 23 - reads better as 'central to understanding'

line 68 - 'for which it had been corrected for'. Please one 'for' is enough - either at the start or end

line 76 - not clear what 'amplitude' means here. Is it the number of species? Perhaps 'magnitude' would better reflect a general idea of size.

line 78 - 'accompanied costs of'. Could you say 'accompanying costs of', or why not simply 'costs of' ?

line 78 - 'predicted by e.g. the'. The e.g. abbreviation doesn't work here. How about 'predicted by, for example, the'

line 85 - 'the residuals of a regression explaining phylogenetic diversity by means of species richness'. Using 'means of' in this statistical context is not ideal because it can also be read as 'averages of'.

How about 'using species richness' or 'based on species richness' ?

This same terminology is used at lines 64, 85, 111, 118, 358 and 446

At line 373, it is used (I think) to actually mean averages.

lines 437-8 - 'that are also occurring' should be 'that also occur'.

line 480 - 'code' when it means computer code, is usually in the singular, representing a type of thing, rather than individual items. So please use 'R code is available' or 'R scripts are available'.

Reviewer #2:

Remarks to the Author:

I think authors did a fantastic effort in reviewing the manuscript, and I'm glad that my previous suggestions (as well as those from reviewer #1) were useful somehow. I have no further comments, just congrats to authors for the job well done!

REVIEWERS REPORTS

Reviewer #1 (Remarks to the Author)

Thank you to the authors, for a very thorough and thoughtful response to issues raised in the first review of this ms. And for making the changes really easy to follow in the revised ms.

The terminology used and rationale for the methods are significantly revised, and the calculation of geographic distance was replaced with a land distance metric. The authors have also been very thorough in testing robustness of the results to choices at many stages in the methods.

The data used throughout the study - true phylogenetic turnover (phylo- β) corrected for geographic distance - are the residuals of a GLM model (logistic regression) for each of the four tetrapod classes. But neither the original phylo- β results nor the results of these models are given.

So can I request one more addition to the results, which would be to include in the main paper the model fit (presumably r^2) for each of the geographic models, and in supplementary a scatterplot of phylo- β for each group with the fitted line plotted. This would give the interested reader a much clearer understanding of the basis of the residual values on which the study is built.

BS et al.: We thank the reviewer #1 for seeing our revised manuscript and the very thorough review and corrections, which again improved and complemented the manuscript. Thank you also for your thorough grammar check, the explanations on grammar-corrections, and the careful figure-caption check. Highly appreciated!

We appreciate that reviewer #1 made us aware of including the original phylo- β results and the correlation between the phylogenetic turnover and both geographic and environmental distance (see rev. #1 comment to lines 211-13 below).

We now include the model fit in the main text (L127-130). In addition, we added the requested scatterplots to the Supplementary. First, Supplementary Fig. 1a shows the here requested scatterplots of the original phylo- β values (Simpson's pairwise dissimilarity index) along geographic land distances and the regression line used to remove the land distance effects. Second, Supplementary Fig. 1b (see rev. #1 request to lines 211-13 below) shows the phylogenetic turnover, which now is corrected for spatial effects, along environmental distances. These scatterplots include the requested correlation statistics.

*** Other suggestions ***

Figure 1:

* why not make the dots and crosses on panels a and b the same colour?

BS et al.: We chose two different colors because there is a difference between black and red dots and crosses. Each single dot or cross in black represents the mean of all pairwise phylo- β values of environmentally neighboring sample sites, while each red dot or cross represents one of the sample sites used to calculate phylo- β between neighboring strata. This was already there in the previous version, but we

agree that it was not explained clearly enough in the caption of Fig. 1. We therefore changed the specific sentence accordingly to make this clearer:

L62–67: “[...] Each black dot or cross refers to the mean of all pairwise phylo- β calculations among sample sites (red dots or crosses in panel **b**) between neighboring strata. **b** Illustration of a hypothesized structuring of phylo- β between neighboring equidistant strata (dotted and solid quadrats) resulting in environmental zones of low (domain A) or high (domain B) values. The mapped phylo- β values (phylo- β (residuals)) represent the means of all pairwise phylo- β calculations among sample sites (red dots or crosses) between neighboring strata. [...]”

BS et al.: While clarifying Figure 1 caption, we changed “sample points” to “sample sites” within the caption and throughout the manuscript to be consistent.

* the y-axis label says '(residuals)' but the colour legend does not. At first I thought the colours represented actual phylo- β (which would add information), rather than duplicating the information given by y-axis position in panel 1. So could you label the colour legend to be clear it is also the residual values?

BS et al.: Thank you for spotting this. We changed the label of the color legend in a) and b) to “Phylo- β (residuals)”.

* in panel c, are NS and niche width the same thing? And in the figure legend.

BS et al.: Thank you, we were not consistent enough in our wording. NS and niche width are the same here. We changed “niche width” to “niche size” in panel c). We did not use the term “niche width” in the legend and therefore had not to change it in the legend. We also changed “niche width” to “niche size” in line 452.

lines 109-10 - 'to what degree phylo- β can be partitioned into geographic and environmental distance effects.' This raises again the intriguing question - how much phylo- β actually was explained by geographic distance??

BS et al.: We now give this information on line 127-130. Thank you for making us aware about these missing data. Geographic distance explained only a small proportion in amphibians and squamates and more in mammals and birds. Although the effect is rather small, it is important to remove this effect when we want to analyze the structure in environmental space. We would like to make the point that our method does not depend on “how much geographic distance actually explains phylo- β ”. We start from the assumption that both geographic and environmental distance explain some, but not necessarily much phylo- β variation (past studies have shown this). Rather, we want to visualize how phylo- β varies along environmental gradients when calculated among equidistant environmental strata. To avoid confounding effects with geographic distance, we simply need to first remove the geographic distance effects, irrespective of how much it actually explains. When adjusting the text, we realized that we used “spatial distance” instead of “geographic distance” in the bold intro text, while we used the latter throughout the text afterwards. We have now unified our wording to “geographic distance”.

L127-132: “**True phylogenetic turnover across geographic land distance.** Geographic land distance explained some of the variation in true phylogenetic turnover (Simpsons pairwise dissimilarity index) between pairs of sample sites (Supplementary Fig. 1a; adjusted regression R^2 for amphibians: 0.03, squamates: 0.04, mammals: 0.29 and birds: 0.46). Hereafter, true phylogenetic turnover (phylo- β) refers to the fraction that is independent of geographic land distance (residuals of Simpsons pairwise dissimilarity index \sim geographic land distance).”

lines 211-13 - 'The discontinuities in phylo- β along environmental gradients are generally congruent with the view that phylo- β is not strongly correlated with environmental distances, a common pattern found in empirical studies'
This seems to be a key finding of the study. Where are the results for the correlation of phylo- β to the two environmental dimensions?

BS et al.: Thank you for making this point. We have added both the correlations (to the results, L133-140) and scatterplots (to the Supplementary Fig. 1b) showing the relationship between phylo- β (residuals) and environmental distance. We also refer to these results on L227-229 in the discussion.

*Results, L133-140: “**True phylogenetic turnover across environmental distance.** Phylo- β varied greatly across environmental distances (Supplementary Fig. 1b). The correlation between phylo- β and environmental distance was significant but relatively low across all clades (amphibians: $r=0.33$, 95% CI [0.327,0.342], $p<0.001$; squamates: $r=0.36$, 95% CI [0.352,0.371], $p<0.001$; mammals: $r=0.29$, 95% CI [0.279,0.294], $p<0.001$; birds: $r=0.35$, 95% CI [0.345,0.359], $p<0.001$). The variation in phylo- β was especially high among short environmental distances, irrespective of the clade analyzed. To visualize the structure in and the drivers behind this variation, we plotted phylo- β along short, equidistant neighboring strata.”*

Discussion, L226-228: “The discontinuities in phylo- β we found along environmental gradients are generally congruent with the view that phylo- β is not strongly correlated with environmental distances, a common pattern found in empirical studies^{5,26,27} and in our data (Supplementary Fig. 1b).”

** Minor wording etc **

line 23 - reads better as 'central to understanding'

BS et al.: We changed it (L23).

line 68 - 'for which it had been corrected for'. Please one 'for' is enough - either at the start or end

BS et al.: We deleted the “for” at the end (L68-69). Thanks for spotting this.

line 76 - not clear what 'amplitude' means here. Is it the number of species? Perhaps 'magnitude' would better reflect a general idea of size.

BS et al.: Yes, that's what we meant. We changed it to “magnitude” (L77).

line 78 - 'accompanied costs of'. Could you say 'accompanying costs of', or why not simply 'costs of' ?

BS et al.: We agree and now simply state “costs of” (L79).

line 78 - 'predicted by e.g. the'. The e.g. abbreviation doesn't work here. How about 'predicted by, for example, the'

BS et al.: We changed this as suggested (L79). Thanks for the correction.

line 85 - 'the residuals of a regression explaining phylogenetic diversity by means of species richness'. Using 'means of' in this statistical context is not ideal because it can also be read as 'averages of'. How about 'using species richness' or 'based on species richness' ?

This same terminology is used at lines 64, 85, 111, 118, 358 and 446

BS et al.: Thank you, this is an important point. We changed it to “based on” at the following lines: L 86, 111, 119, 373 and 461.

We did not change it at line 64 (now L66), because this sentence refers to statistical mean values (averages):

L66: “The mapped phylo- β values (phylo- β (residuals)) represent the means of all pairwise phylo- β calculations among sample sites (red dots or crosses) between neighboring strata.”

At line 373, it is used (I think) to actually mean averages.

BS et al.: Yes, that’s correct.

lines 437-8 - 'that are also occurring' should be 'that also occur'.

BS et al.: Changed (L452).

line 480 - 'code' when it means computer code, is usually in the singular, representing a type of thing, rather than individual items. So please use 'R code is available' or 'R scripts are available'.

BS et al.: We highly appreciate the thorough grammar check and careful explanations/suggestions. We changed it to “R code is available” (L495).

Reviewer #2 (Remarks to the Author):

I think authors did a fantastic effort in reviewing the manuscript, and I'm glad that my previous suggestions (as well as those from reviewer #1) were useful somehow. I have no further comments, just congrats to authors for the job well done!

BS et al.: Thanks a lot for this positive feedback and your careful check on our revised manuscript.

Reviewers' Comments:

Reviewer #1:

Remarks to the Author:

Thanks to the authors for dealing with these remaining issues. They have all been resolved, and it is a great paper!

Only 2 trivial suggestions:

Figure 1 - the beta symbols appear as !

Line 140 - as this refers to plotting, should there be a figure reference?

REVIEWERS REPORTS

Reviewer #1 (Remarks to the Author):

Thanks to the authors for dealing with these remaining issues. They have all been resolved, and it is a great paper!

BS et al.: We thank the reviewer #1 for seeing our revised manuscript and the compliments .

Only 2 trivial suggestions:

Figure 1 - the beta symbols appear as !

BS et al.: We changed all beta symbols in Figure 1-3. Apparently, they were written in the less common font "Cambria Math" and therefore appeared as "!" on computers lacking this font. We changed it to the required "Symbol" font.

We also realized that we sometimes used "phylo β " or "phylo- β ". We now consistently use "phylo- β " throughout the main text and the Supplementary.

Line 140 - as this refers to plotting, should there be a figure reference?

BS et al.: We now refer to Figure 2 in this sentence.

L163: "To visualize the structure in and the drivers behind this variation, we plotted phylo- β along short, equidistant neighboring strata (Fig. 2)."